# A Complete Characterization of Learnability for Stochastic Noisy Bandits

**Steve Hanneke**                                                        STEVE.HANNEKE@GMAIL.COM
*Purdue University*

**Kun Wang**                                                            WANGKUN8512@GMAIL.COM
*Purdue University*

**Editors:** Gautam Kamath and Po-Ling Loh

## Abstract

We study the stochastic noisy bandit problem with an unknown reward function $f^*$ in a known function class $\mathcal{F}$. Formally, a model $M$ maps arms $\pi$ to a probability distribution $M(\pi)$ of reward. A model class $\mathcal{M}$ is a collection of models. For each model $M$, define its mean reward function $f^M(\pi) = \mathbb{E}_{r \sim M(\pi)}[r]$. In the bandit learning problem, we proceed in rounds, pulling one arm $\pi$ each round and observing a reward sampled from $M(\pi)$. With knowledge of $\mathcal{M}$, supposing that the true model $M \in \mathcal{M}$, the objective is to identify an arm $\hat{\pi}$ of near-maximal mean reward $f^M(\hat{\pi})$ with high probability in a bounded number of rounds. If this is possible, then the model class is said to be learnable.

Importantly, a result of Hanneke and Yang (2023) shows there exist model classes for which learnability is undecidable. However, the model class they consider features deterministic rewards, and they raise the question of whether learnability is decidable for classes containing sufficiently noisy models. More formally, for any function class $\mathcal{F}$ of mean reward functions, we denote by $\mathcal{M}_\mathcal{F}$ the set of all models $M$ such that $f^M \in \mathcal{F}$. In other words, $\mathcal{M}_\mathcal{F}$ admits arbitrary zero-mean noise. Hanneke and Yang (2023) ask the question: Can one give a simple complete characterization of which function classes $\mathcal{F}$ satisfy that $\mathcal{M}_\mathcal{F}$ is learnable?

For the first time, we answer this question in the positive by giving a complete characterization of learnability for model classes $\mathcal{M}_\mathcal{F}$. In addition to that, we also describe the full spectrum of possible optimal query complexities. Further, we prove adaptivity is sometimes necessary to achieve the optimal query complexity. Last, we revisit an important complexity measure for interactive decision making, the Decision-Estimation-Coefficient (Foster et al., 2021, 2023), and propose a new variant of the DEC which also characterizes learnability in this setting.

**Keywords:** Bandits, Structured Bandits, Learning Theory, Query Complexity

## 1. Introduction

The multi-armed bandit problem (Robbins, 1952; Auer et al., 2002a,b; Lattimore and Szepesvári, 2020) is a problem in which a learner performs an action and gains a reward round by round, with the intention of identifying actions with the highest rewards. The multi-armed bandit problem occurs in many contexts. For instance, imagine one situation where a restaurant customer wants to figure out which item on the menu is the most delicious. By strategically choosing his order each time he visits the restaurant, he can eventually identify which item is the most delicious. Many other examples in practice include recommendation systems, clinical trials, and financial portfolio design (Yue and Joachims, 2009; Combes et al., 2015; Li et al., 2016, 2010; Slivkins, 2011a). The main theoretical interests are trying to identify an optimal strategy and corresponding optimal query complexity.

From a theoretical perspective, one important recent line of work explores the role of *model class* in the multi-armed bandit problem. The framework of bandit learning with model classes is analogous to the idea of concept classes in the PAC learning framework (Vapnik and Chervonenkis, 1974; Valiant, 1984). The PAC learning framework provides a concise and elegant characterization of learnability and query complexity based on the concept class, thereby unifying different learning problems into a single theory. In contrast, most of the existing bandit literature focuses on different special cases, such as linear bandits (McMahan and Blum, 2004; Awerbuch and Kleinberg, 2008; Abbasi-Yadkori et al., 2011), Lipschitz bandits (Agrawal, 1995; Kleinberg, 2004; Kleinberg et al., 2008; Slivkins, 2011b) and bandits with smooth functions on a metric space (Bubeck et al., 2011). There is no unified theory in the bandit literature. In the context of the multi-armed bandit problem, a model maps actions (also known as arms) to reward distributions, and a model class is a set of models. The query complexity is the minimal number of queries enough to identify the arm with near-maximal mean reward. If query complexity is finite, then we say the model class is learnable. In this setting, one natural question would be: Is it possible to give a general theory based on the model class in the multi-armed bandit problem? Several papers have already made some efforts in this direction. Amin et al. (2011) study exact learning in bandit problem, proposing a corresponding Haystack Dimension. Foster et al. (2021, 2023) give a lower bound and a non-matching upper bound for the query complexity of the multi-armed bandit problem based on their proposed complexity measure, the Decision-Estimation-Coefficient. Hanneke and Yang (2023) identify the optimal query complexity of deterministic rewards for binary-valued bandits. Other works have studied the problem of bandit learning with model classes under the name of structured bandits since a long time ago (Combes et al., 2017; Tirinzoni et al., 2020; Van Parys and Golrezaei, 2024). However, a complete characterization of learnability in general has remained elusive.

Perhaps surprisingly, recent work shows this is unavoidable: a result of Hanneke and Yang (2023) reveals that there exist model classes for which learnability is undecidable within ZFC. Nevertheless, one promising direction would be identifying fairly general subfamilies of model classes for which there exist complete characterizations of learnability. In addition, the model class Hanneke and Yang (2023) consider features deterministic reward. However in the bandit literature, most of existing works admit noisy rewards structure. Therefore, Hanneke and Yang (2023) ask the following question: does there exist a simple and complete characterization of learnability for bandits with arbitrary (zero-mean) noise?

Formally, we define the stochastic noisy bandit problem as follows. There is a set of arms $\Pi$. A model $M$ maps arms $\pi$ to reward distributions $M(\pi)$. A model class $\mathcal{M}$ is a collection of models. For each model $M$, its mean reward function $f^M(\pi) = \mathbb{E}_{r \sim M(\pi)}[r] \in [0, 1]$. In other words, for a model $M$, each arm $\pi \in \Pi$ has a reward distribution $M(\pi)$ with mean value $f^M(\pi)$. Our main interest in this paper is model classes induced by *function classes*. A function class $\mathcal{F}$ is defined as a collection of functions $f : \Pi \to [0, 1]$. For any function class $\mathcal{F}$, we denote by $\mathcal{M}_{\mathcal{F}}$ the set of all models $M$ such that $f^M \in \mathcal{F}$. In other words, $\mathcal{M}_{\mathcal{F}}$ admits models whose mean function is in $\mathcal{F}$, but allows for arbitrary zero-mean noise. The learning problem induced by model class $\mathcal{M}_{\mathcal{F}}$ is described as follows: it proceeds round by round. In each round $i$, the learner chooses one arm $\pi_i$ to pull and receives a reward $r(\pi_i) \in [0, 1]$, which is a random variable sampled from $M(\pi_i)$ (conditionally independent of the past given $\pi_i$). With the knowledge of $\mathcal{F}$, suppose the true model $M \in \mathcal{M}_{\mathcal{F}}$, the objective of the learner is to identify an arm $\hat{\pi}$ such that $f^M(\hat{\pi}) \geq \sup_{\pi} f^M(\pi) - \alpha$ with probability at least $1 - \delta$ in a bounded number of rounds, $\forall \alpha, \delta \in (0, 1)$ and $\forall M \in \mathcal{M}_{\mathcal{F}}$. If this is possible, we say function class $\mathcal{F}$ is learnable with arbitrary noise. The minimal bound

on the number of rounds sufficient for achieving this is called the query complexity, denoted by $QC(\mathcal{F}, \alpha, \delta)$.

In this work, for the first time, we give a complete characterization of learnability for stochastic noisy bandits with arbitrary noise: Define *generalized maximin volume* of function class $\mathcal{F}$

$$\gamma_{\mathcal{F},\alpha} = \sup_{p \in \Delta(\Pi)} \inf_{f \in \mathcal{F}} \mathbb{P}_{\pi \sim p}\left(\sup_{\pi^*} f(\pi^*) - f(\pi) \leq \alpha\right), \tag{1}$$

where $\Delta(\Pi)$ is the set of all distributions on $\Pi$. Then we have:

**Theorem 1** *$\mathcal{F}$ is learnable with arbitrary noise if and only if $\gamma_{\mathcal{F},\alpha} > 0 \;\forall \alpha \in (0, 1)$.*

This establishes the first complete characterization of learnability for stochastic noisy bandits, which resolves the open question proposed by Hanneke and Yang (2023), and completes a long line of research on bandit learnability which has been studied many years[1]. In addition, we also have several other results:

- We further explore the optimal query complexity of stochastic noisy bandits and discover it can range from $\tilde{\Theta}\left(\log \frac{1}{\gamma_{\mathcal{F},\alpha}}\right)$ to $\tilde{\Theta}\left(\frac{1}{\gamma_{\mathcal{F},\alpha}}\right)$.[2]

- We also find adaptivity is sometimes necessary for achieving the optimal query complexity.

- We extend arbitrary bounded noise to arbitrary unbounded noise using the median of means method.

- We find the $\Omega\left(\log \frac{1}{\gamma_{\mathcal{F},\alpha}}\right)$ lower bound still remains valid when the model class $\mathcal{M}$ contains only Gaussian noise.

- We propose a variant of the well-known Decision-Estimation-Coefficient (Foster et al., 2021, 2023) and prove it can also characterize learnability of stochastic bandits with arbitrary noise.

We organize our paper as follows: In Section 2, we provide our main learnability characterization: we present the upper and lower bounds on the query complexity based on *generalized maximin volume* and extend learnability result to no-regret setting. Section 3 describes two results on the optimal query complexity. Section 4 extends our learnability result to more general noise settings. Section 5 presents our variant of the Decision-Estimation-Coefficient with corresponding query complexity analysis.

## 2. Characterization of Learnability

In this section, we introduce our main learnability results. Theorem 2 and Theorem 3 give sufficient and necessary conditions of learnability and corresponding query complexities, respectively. Combining Theorem 2 and Theorem 3 gives our characterization of learnability (Theorem 1). Finally, we extend our learnability results to no-regret learnability (Theorem 6) through a equivalence between two settings.

---

1. The work studying the general bandit learnability problem can be traced back to at least Amin et al. (2011), though work on stochastic bandits even dates back to Robbins (1952).
2. We use $\tilde{\Theta}(f)$ or $\tilde{O}(f)$ to hide additional polylog factor of function $f$.

---

**Algorithm 1** Learning algorithm for function class $\mathcal{F}$

---

**Input:** Parameter $\alpha$, $\delta$

Recall Equation (1), $\gamma_{\mathcal{F},\alpha/2} = \sup_p \inf_{f \in \mathcal{F}} \mathbb{P}_{\pi \sim p} \left( \sup_{\pi^*} f(\pi^*) - f(\pi) \leq \alpha/2 \right)$.

Let $p$ be the distribution that achieve $\gamma_{\mathcal{F},\alpha/2}$.

Sample $m = \frac{1}{\gamma_{\mathcal{F},\alpha/2}} \log \frac{2}{\delta}$ arms from $p : \pi_1, \pi_2, ..., \pi_m$.

For each $\pi_i$, query $\frac{8}{\alpha^2} \log \frac{4m}{\delta}$ times and let $\hat{f}(\pi_i)$ denote the empirical mean of these rewards.

**Output:** The arm $\hat{\pi}$ with the largest empirical mean $\hat{f}(\hat{\pi})$.

---

**Theorem 2 (Upper bound)** $\gamma_{\mathcal{F},\alpha} > 0 \ \forall \alpha \in (0,1)$ *is a sufficient condition for learnability of $\mathcal{F}$ with arbitrary noise. In addition,* $QC(\mathcal{F}, \alpha, \delta) = O \left( \frac{8}{\gamma_{\mathcal{F},\alpha/2}\alpha^2} \log \frac{2}{\delta} \log \left( \frac{4}{\delta \gamma_{\mathcal{F},\alpha/2}} \log \frac{2}{\delta} \right) \right)$.

**Proof** Recall that $\gamma_{\mathcal{F},\alpha/2} = \inf_{f \in \mathcal{F}} \mathbb{P}_{\pi \sim p} \left( \sup_{\pi^*} f(\pi^*) - f(\pi) \leq \alpha/2 \right)$ and $p$ is the distribution that achieves $\gamma_{\mathcal{F},\alpha}$. Based on the definition of $p$, we have $\mathbb{P}_{\pi \sim p} \left( \sup_{\pi^*} f(\pi^*) - f(\pi) \leq \alpha/2 \right) \geq \gamma_{\mathcal{F},\alpha/2} > 0 \ \forall f \in \mathcal{F}$. When we sample $\pi$ from distribution $p$ for $m = \frac{1}{\gamma_{\mathcal{F},\alpha/2}} \log(\frac{2}{\delta})$ times, we have:

$$\mathbb{P} \left( \exists \pi_i : \sup_{\pi^*} f(\pi^*) - f(\pi_i) \leq \alpha/2 \right)$$
$$= 1 - \mathbb{P} \left( \forall \pi_i : \sup_{\pi^*} f(\pi^*) - f(\pi_i) > \alpha/2 \right)$$
$$\geq 1 - (1 - \gamma_{\mathcal{F},\alpha/2})^{\frac{1}{\gamma_{\mathcal{F},\alpha/2}} \log(\frac{2}{\delta})} \tag{2}$$
$$\geq 1 - e^{-\log(\frac{2}{\delta})}$$
$$= 1 - \frac{\delta}{2}.$$

Therefore, with probability at least $1 - \frac{\delta}{2}$, $\exists \pi_i, \sup_{\pi^*} f(\pi^*) - f(\pi_i) \leq \alpha/2$. Then, based on Lemma 18, let $\hat{f}(\pi_i)$ be the empirical mean of these rewards in $\frac{8}{\alpha^2} \log \frac{4m}{\delta}$ rounds,

$$\mathbb{P} \left[ \left| f(\pi_i) - \hat{f}(\pi_i) \right| \geq \frac{\alpha}{4} \right] \leq 2 e^{-2 \frac{8}{\alpha^2} \log \frac{4m}{\delta} \frac{\alpha^2}{16}} = \frac{\delta}{2m}.$$

Using the union bound, with probability at least $1 - \frac{\delta}{2}$, the empirical mean of every arm $\pi_i$ lies within $\frac{\alpha}{4}$ of its true mean. Consequently, with probability at least $1 - \delta$, the arm $\hat{\pi}$ with the largest estimated mean satisfies $\sup_{\pi^*} f(\pi^*) - \hat{f}(\hat{\pi}) \leq \frac{3}{4}\alpha$. Thus, for the arm $\hat{\pi}$, it follows that $\sup_{\pi^*} f(\pi^*) - f(\hat{\pi}) \leq \alpha$, as desired.

$\blacksquare$

**Theorem 3 (Lower bound)** $\gamma_{\mathcal{F},\alpha} > 0 \ \forall \alpha \in (0,1)$ *is a necessary condition for learnability of $\mathcal{F}$ with arbitrary noise. In addition,* $QC(\mathcal{F}, \alpha, \delta) = \Omega \left( \log \frac{1}{\gamma_{\mathcal{F},\alpha}} \right)$.

**Proof** We want to show if a function class $\mathcal{F}$ is learnable, then it must be $\gamma_{\mathcal{F},\alpha} > 0 \ \forall \alpha \in (0,1)$. In this proof, we will focus on models with binary-valued noisy rewards. First, we have $\forall M^* \in \mathcal{M}_{\mathcal{F}}, \exists M_B^* \in \mathcal{M}_{\mathcal{F}}$, s.t. $f^{M_B^*} = f^{M^*}$ and $M_B^*(\pi)$ is supported on $\{0,1\} \ \forall \pi$. Supposing $\mathcal{F}$ is

learnable, let $A$ be any learning algorithm for $\mathcal{F}$ and let $T$ be the corresponding query complexity for a given $\alpha$ and $\delta$. $\forall M^* \in \mathcal{M}_\mathcal{F}$, if we simulate running $A$ under $M_B^*$, then with probability at least $1 - \delta$, it returns $\hat{\pi}$ such that $\sup_{\pi^*} f^{M^*}(\pi^*) - f^{M^*}(\hat{\pi}) \le \alpha$.

We construct a distribution $p$ that witnesses $\gamma_{\mathcal{F},\alpha}$ is lower bounded by a function of $T$. We execute the algorithm $A$, but whenever it pulls an arm, we respond with an independent Bernoulli($\frac{1}{2}$) reward. Let $p$ be the distribution over the $\hat{\pi}$ output by this execution. Then $\forall M^* \in \mathcal{M}_\mathcal{F}$, with probability $2^{-T}$, the Bernoulli($\frac{1}{2}$) rewards all agree with the rewards $M_B^*$ would respond with, and independent of which rewards $M_B^*$ would respond with. Thus, we have $\mathbb{P}_{\pi \sim p}(\sup_{\pi^*} f^{M^*}(\pi^*) - f^{M^*}(\pi) \le \alpha) \ge (1-\delta)2^{-T} > 0$.

Finally, we have $\gamma_{\mathcal{F},\alpha} \ge \mathbb{P}_{\pi \sim p}(\sup_{\pi^*} f^{M^*}(\pi^*) - f^{M^*}(\pi) \le \alpha) \ge (1-\delta)2^{-T} > 0$. In addition, we have: $T \ge \log \frac{1-\delta}{\gamma_{\mathcal{F},\alpha}} = \Omega\left(\log \frac{1}{\gamma_{\mathcal{F},\alpha}}\right)$.

$\blacksquare$

**Remark 4** *Since our lower bound proof is based on binary noise, this indicates $\gamma_{\mathcal{F},\alpha} > 0 \ \forall \alpha \in (0,1)$ is also characterization of learnability for $\mathcal{F}$ with binary noise (when the rewards of $\pi_i$ are binary-valued).*

Next in this section, we extends our learnability result to no-regret setting. We say a function class $\mathcal{F}$ is no-regret learnable with arbitrary noise in the stochastic bandit setting if there is an algorithm $A$ and a function $\mathcal{R} : \mathbb{N} \to [0, \infty)$ with $\mathcal{R}(T) = o(T)$ such that, for any $M^* \in \mathcal{M}_\mathcal{F}$ and any $T \in \mathbb{N}$, $T \sup_x f^{M^*}(x) - \mathbb{E}[\sum_{i=1}^T r(\pi_i)] \le \mathcal{R}(T)$. Now, we restate an equivalence result between our setting and no-regret setting in terms of learnability (Hanneke and Yang, 2023). (While Hanneke and Yang (2023) focuses on the noiseless setting, their proof still applies to noisy settings.)

**Theorem 5 (Hanneke and Yang (2023), Theorem 2 )** *For any noise model, any $(\Pi, \mathcal{F})$ is learnable in the bandit setting if and only if it is no-regret learnable in the bandit setting.*

Combining Theorem 1 and Theorem 5 gives our no-regret learnability results (Theorem 6).

**Theorem 6** *$\mathcal{F}$ is no-regret learnable with arbitrary noise if and only if $\gamma_{\mathcal{F},\alpha} > 0 \ \forall \alpha \in (0,1)$.*

Finally, we give some illustrating examples, showing how to calculate *generalized maximin volume* in these cases.

**Example 1 ($K$-armed bandit)** *Consider any finite $\Pi$ and $\mathcal{F} = [0,1]^\Pi$ , the set of all functions $\Pi \to [0,1]$. In this case, choose distribution $p$ to be uniform over $|\Pi|$ arms, we have $\gamma_{\mathcal{F},\alpha} = \frac{1}{|\Pi|}$.*

**Example 2 (Linear bandit)** *Consider $\Pi = \mathbb{S}^d$, the origin-centered unit ball in $\mathbb{R}^d$ for some $d \in \mathbb{N}$, and $\mathcal{F} = \{x \to w^\top x : w \in \mathbb{S}^d\}$. Consider $d$ is a constant, $\gamma_{\mathcal{F},\alpha} = \Omega(\alpha^d)$. We may construct an $\alpha$-net over the arm space $\Pi$ based on 2-norm. The size of this $\alpha$-net is $O((\frac{1}{\alpha})^d)$. For any $w \in \mathcal{F}$, let $\pi^* = \arg\max_{\pi \in \Pi} w^\top \pi$, let $\pi$ be the closet element of the $\alpha$-net. We have $w^\top \pi^* - w^\top \pi \le \|w\|\|\pi^* - \pi\| \le 1 \cdot \alpha \le \alpha$ based on Cauchy–Schwarz inequality. Let the distribution $p$ be uniform over those net elements, thus we have $\gamma_{\mathcal{F},\alpha} = \Omega(\alpha^d)$.*

**Example 3 (Singletons)** *Consider $\Pi = \mathbb{N}$ and $\mathcal{F} = \{\mathbb{I}_{\{\pi\}}, \pi \in \Pi\}$ : the class of singletons. This is a not learnable class. In this case, $\gamma_{\mathcal{F},\alpha} = 0$ since for any distribution $p$, there exists some arm $\pi \in \Pi$ with low probability mass. Consider the function $f$ such that $f(\pi) = 1$, we have $\mathbb{P}_{\pi \in p}(\sup_{\pi^*} f(\pi^*) - f(\pi) \le \alpha) = 0$.*

## 3. Optimal Query Complexity

Further in this section, we explore some properties of optimal query complexity. Theorem 7 shows every query complexity between $\tilde{\Theta}\left(\log\frac{1}{\gamma_{\mathcal{F},\alpha}}\right)$ and $\tilde{\Theta}\left(\frac{1}{\gamma_{\mathcal{F},\alpha}}\right)$ is achievable. Theorem 8 shows adaptivity is sometimes necessary for achieving optimal query complexity.

**Theorem 7** *For $\alpha \in (0, \frac{1}{3})$ and $\delta \in (0, \frac{1}{2})$, $\forall \gamma \in \{\frac{1}{i} : i \in \mathbb{N}\}$, $\forall f(\gamma) \in \left[\log\frac{1}{\gamma}, \frac{1}{\gamma}\right]$, there exists a function class $\mathcal{F}$ such that $\gamma_{\mathcal{F},\alpha} = \gamma$ and $QC(\mathcal{F}, \alpha, \delta) = \tilde{\Theta}(f(\gamma_{\mathcal{F},\alpha}))$.*

**Proof** Let $\alpha \in (0, \frac{1}{3})$ and $N \in \mathbb{N}$. Consider the function class $\mathcal{F}$ that admits the following binary tree structure: In this tree, each internal node corresponds to an arm, each leaf node corresponds to a bucket of $N$ arms and each edge has a value. The edge pointing to the left child has value $\frac{1}{3}$. The edge pointing to the right child has value $\frac{2}{3}$. The arms inside the buckets either have a mean value of $0$ or $1$. In total, it has $\left\lceil\frac{1}{\gamma_{\mathcal{F},\alpha}N}\right\rceil$ leaves. The height of this tree is thus $\log\left\lceil\frac{1}{\gamma_{\mathcal{F},\alpha}N}\right\rceil$. Each function $f$ in the function class $\mathcal{F}$ is defined as follows: it has only one optimal arm with mean value $1$, which corresponds to an arm in some bucket. For the internal nodes in the branch from the root to this bucket, if the edge in the branch points to the left child, then the corresponding arm in $f$ has mean value $\frac{1}{3}$; if the edge in the branch points to the right child, then the corresponding arm in $f$ has mean value $\frac{2}{3}$. For the nodes off-branch, they all have a mean value $0$. For the arms in the buckets except for the optimal one, they also have mean value $0$.

First, we will argue $QC(\mathcal{F}, \alpha, \delta) \leq \tilde{O}\left(\log\frac{1}{\gamma_{\mathcal{F},\alpha}} + N\right)$. The algorithm can be designed based on the tree structure: Begin by querying the arm corresponding to the root of the tree. Query it sufficiently many times so that one can guarantee the mean of the arm belongs to $\frac{1}{3}$ or $\frac{2}{3}$ with high probability. Follow the branch consistent with the value of the querying result. Repeat this procedure for each subsequent internal node along the branch until reaching a leaf node containing a bucket of arms. Then query every node in this bucket enough times and choose the optimal one. This process ensures $QC(\mathcal{F}, \alpha, \delta) \leq \tilde{O}\left(\log\frac{1}{\gamma_{\mathcal{F},\alpha}} + N\right)$.

Next, we will argue that identifying a near-optimal arm within this function class $\mathcal{F}$ needs at least querying $\Omega\left(\max\left\{\log\left(\frac{1}{\gamma_{\mathcal{F},\alpha}}\right), \frac{N}{2}\right\}\right)$ times. For the lower bound proof, we consider the model class without noise. First, we will show $QC(\mathcal{F}, \alpha, \delta) \geq \frac{N}{2}$. For every function $f$ in the function class $\mathcal{F}$, there exists a target bucket of size $N$ that contains the optimal arm. Let $\pi^* \sim \text{Uniform}(1, ..., N)$. Let $A$ be any learning algorithm and let $\pi_1, ..., \pi_t$ be the sequence of arms it would pull if every reward it receives is $0$ and let $\hat{\pi}$ be its returned arm if all of its rewards received is $0$. In the true scenario, if $\pi^* \notin \{\pi_1, ..., \pi_t, \hat{\pi}\}$, the algorithm will pull $\pi_1, ..., \pi_t$ and output $\hat{\pi}$. Therefore, it fails. $\mathbb{P}(\pi^* \notin \{\pi_1, ..., \pi_t, \hat{\pi}\}) \geq \frac{N-t-1}{N}$. If $t \leq \frac{N}{2} - 1$, then $\frac{N-t-1}{N} \geq \frac{1}{2} > \delta$ if we take $\delta \in (0, \frac{1}{2})$. Then, $\max_{\pi^*} \mathbb{P}(\pi^* \notin \{\pi_1, ..., \pi_t, \hat{\pi}\}) \geq \mathbb{E}_{\pi^* \sim \text{Uniform}(1,...,N)}[\mathbb{P}(\pi^* \notin \{\pi_1, ..., \pi_t, \hat{\pi}\}|\pi^*)] = \mathbb{P}(\pi^* \notin \{\pi_1, ..., \pi_t, \hat{\pi}\}) > \delta$. Therefore, $t \geq \frac{N}{2}$.

Then, we will show $QC(\mathcal{F}, \alpha, \delta) \geq \Omega\left(\log\frac{1}{\gamma_{\mathcal{F},\alpha}}\right)$. Let $N = 1$, then there are $\left\lceil\frac{1}{\gamma_{\mathcal{F},\alpha}}\right\rceil$ leaves, thus the tree has depth $\log\left(\left\lceil\frac{1}{\gamma_{\mathcal{F},\alpha}}\right\rceil\right)$. This can be viewed as active learning with membership query problem.[3] From this perspective, we are able to use Theorem 1 in Kulkarni et al. (1993)

---

3. In Kulkarni et al. (1993), their problem allows for $\epsilon$ error in the objective. However, our setting requires an exact function. Thus we need to set $\epsilon$ to be $0$. In our setting, based on our construction, each time we query arms, it is equivalent to asking whether the target function is within some set.

to get a lower bound in our setting: $\Omega\left(\log\frac{1}{\gamma_{\mathcal{F},\alpha}}\right)$ queries is necessary for our setting. Therefore, $QC(\mathcal{F},\alpha,\delta) \geq \Omega\left(\log\frac{1}{\gamma_{\mathcal{F},\alpha}}\right)$.

Since the number of arms each buckets $N$ can range from 1 to $\frac{1}{\gamma_{\mathcal{F},\alpha}}$. So $QC(\mathcal{F},\alpha,\delta)$ can range from $\tilde{\Theta}\left(\log\frac{1}{\gamma_{\mathcal{F},\alpha}}\right)$ to $\tilde{\Theta}\left(\frac{1}{\gamma_{\mathcal{F},\alpha}}\right)$. ∎

The query of the adaptive algorithm each round might depend on the previous queries and corresponding rewards. On the contrary, the query of the non-adaptive algorithm is determined in advance before all rounds start. Our lower bound proof for learnability uses the non-adaptive algorithm. It is interesting to know that the adaptive algorithm improves the query complexity in some cases. Formally, we give our result Theorem 8.

**Theorem 8** $\forall\gamma \in \{2^{-i} : i \in \mathbb{N}\}$, *there exists a function class $\mathcal{F}$ such that $\gamma_{\mathcal{F},\alpha} = \gamma$ and for this function class $\mathcal{F}$, there exists an adaptive algorithm with $\tilde{O}\left(\log\frac{1}{\gamma_{\mathcal{F},\alpha}}\right)$ query complexity and every non-adaptive algorithm has query complexity $\Omega\left(\frac{1}{\gamma_{\mathcal{F},\alpha}}\right)$.*

**Proof** In this proof, we use the same function class construction $\mathcal{F}$ as in the proof of Theorem 7, with $N = 1$. The upper bound proof follows the same reasoning as in Theorem 7. The remaining task is to establish the $\Omega\left(\frac{1}{\gamma_{\mathcal{F},\alpha}}\right)$ lower bound for non-adaptive algorithms. We continue to analyze the model class without noise. Let $A$ denote the algorithm, and let the version space represent the set of all functions in the function class that are consistent with the existing observations. Denote the arm selected by the algorithm as $\hat{\pi}$ and the optimal arm as $\pi^*$. Our goal is to lower bound $\min_A \max_{\pi^*} \mathbb{P}_{\pi^*}(\hat{\pi} \neq \pi^*)$. Let $\boldsymbol{\pi^*} \sim \text{Uniform(leaves)}$, then we have:

$$
\begin{aligned}
&\min_A \max_{\pi^*} \mathbb{P}_{\pi^*}(\hat{\pi} \neq \pi^*) \\
&\geq \min_A \mathbb{E}[\mathbb{P}_{\pi^*}(\hat{\pi} \neq \pi^*|\pi^*)] \\
&= \min_A \mathbb{P}(\hat{\pi} \neq \boldsymbol{\pi^*}) \\
&= \min_A \mathbb{E}[\mathbb{P}(\hat{\pi} \neq \boldsymbol{\pi^*}|\pi_1, f^*(\pi_1), ..., \pi_n, f^*(\pi_n), \hat{\pi})].
\end{aligned}
\tag{3}
$$

The algorithm has some fixed distribution $q$. Since it is non-adaptive, it only observes $n$ samples satisfying $(\pi_1, ..., \pi_n) \sim q$ (not necessarily i.i.d.). Then we have:

$$
\mathbb{P}(\hat{\pi} \neq \boldsymbol{\pi^*}|\pi_1, f^*(\pi_1), ..., \pi_n, f^*(\pi_n), \hat{\pi}) \geq 1 - \frac{1}{N(\pi_1, ..., \pi_n, \boldsymbol{\pi^*})},
$$

where $N(\pi_1, ..., \pi_n, \boldsymbol{\pi^*})$ is the number of leaves in the version space within the tree structure. Therefore,

$$
\begin{aligned}
&\mathbb{P}(\hat{\pi} \neq \boldsymbol{\pi^*}) \\
&= \mathbb{E}[\mathbb{P}(\hat{\pi} \neq \boldsymbol{\pi^*}|\pi_1, f^*(\pi_1), ..., \pi_n, f^*(\pi_n), \hat{\pi})] \\
&\geq 1 - \mathbb{E}\left[\frac{1}{N(\pi_1, ..., \pi_n, \boldsymbol{\pi^*})}\right] \\
&= 1 - \mathbb{E}\left[\mathbb{E}\left[\left.\frac{1}{N(\pi_1, ..., \pi_n, \boldsymbol{\pi^*})}\right| \pi_1, ..., \pi_n\right]\right].
\end{aligned}
\tag{4}
$$

Then we have the following observation: If $\boldsymbol{\pi^*}$, $\boldsymbol{\pi^*}$'s sibling, and $\boldsymbol{\pi^*}$'s parent are not included in $\{\pi_1, \ldots, \pi_n\}$, then both $\boldsymbol{\pi^*}$ and its sibling remain in the version space, implying that $N(\pi_1, \ldots, \pi_n, \boldsymbol{\pi^*}) \geq 2$.

Let $n = \frac{1}{10\gamma}$. There exists at least $\frac{1}{2\gamma} - n = \frac{2}{5\gamma}$ parent level nodes such that there does not exists $\pi_i$ among that parent or either child. This implies with probability greater than $\frac{2/(5\gamma)}{1/(2\gamma)} = \frac{4}{5}$, $\boldsymbol{\pi^*}$ is a child of such a node, thus $N(\pi_1, ..., \pi_n, \boldsymbol{\pi^*}) \geq 2$. $\mathbb{E}\left[\frac{1}{N(\pi_1, ..., \pi_n, \boldsymbol{\pi^*})} \middle| \pi_1, ..., \pi_n\right] \leq \frac{1}{10} + \frac{4}{5} \cdot \frac{1}{2} = \frac{1}{2}$. Finally, we conclude $\mathbb{P}(\hat{\pi} \neq \boldsymbol{\pi^*}) \geq 1 - \frac{1}{2} = \frac{1}{2}$. Choosing $\delta \in (0, \frac{1}{2})$ completes the lower bound proof. ∎

## 4. Extension to Unbounded and Gaussian Noise

In this section, we extend our existing results to broader and well-known scenarios. First, consider the model $M$ such that for each arm $\pi$, its reward $r(\pi)$ sampled from $M(\pi)$ is unbounded but with variance $\sigma^2$. Let $\mathcal{M}_{\mathcal{F}}^{U,\sigma^2}$ denote a set of all such models such that $f \in \mathcal{F}$. With the knowledge of $\mathcal{F}$, suppose the true model $M \in \mathcal{M}_{\mathcal{F}}^{U,\sigma^2}$, if there exists an algorithm that is able to identify an arm $\hat{\pi}$ such that $f^M(\hat{\pi}) \geq \sup_\pi f^M(\pi) - \alpha$ with probability at least $1 - \delta$ in a bounded number of rounds $\forall \alpha, \delta \in (0, 1)$ and $\forall M \in \mathcal{M}_{\mathcal{F}}^{U,\sigma^2}$, then we say function class $\mathcal{F}$ is learnable with unbounded noise.

Theorem 9 demonstrates that Algorithm 2 can identify a near-optimal arm in a finite number of rounds for $\mathcal{M}_{\mathcal{F}}^{U,\sigma^2}$, whenever such identification is possible. In essence, Algorithm 2 replaces direct mean estimation with the median-of-means method, which offers improved accuracy guarantees when rewards are unbounded.

---

**Algorithm 2** Learning algorithm for function class $\mathcal{F}$ with unbounded reward

---

**Input:** Parameter $\alpha$, $\delta$, $\sigma$, $c_M$ (constant in median-of-mean method)
Recall Equation (1), $\gamma_{\mathcal{F},\alpha/2} = \sup_p \inf_{f \in \mathcal{F}} \mathbb{P}_{\pi \sim p}(\sup_{\pi^*} f(\pi^*) - f(\pi) \leq \alpha/2)$.
Let $p$ be the distribution that achieves $\gamma_{\mathcal{F},\alpha/2}$.
Sample $m = \frac{1}{\gamma_{\mathcal{F},\alpha/2}} \log \frac{2}{\delta}$ arms from $p : \pi_1, \pi_2, ..., \pi_m$.
For each $\pi_i$, query $\frac{16 c_M \sigma^2 \log \frac{2m}{\delta}}{\alpha^2}$ times and estimate mean $\kappa$ using median of mean:
Evenly partition the data into $\log \frac{1}{\delta}$ groups and let $\kappa$ be the median of the set of means of the groups.
**Output:** The arm $\hat{\pi}$ with the largest estimated mean.

---

**Theorem 9** $\gamma_{\mathcal{F},\alpha} > 0 \ \forall \alpha \in (0, 1)$ *is a sufficient condition for learnability of $\mathcal{F}$ with unbounded noise.*

**Proof** First, same as analysis of Theorem 2, with probability at least $1 - \frac{\delta}{2}$, $\exists \pi_i, \sup_{\pi^*} f(\pi^*) - f(\pi_i) \leq \frac{\alpha}{2}$. Based on Lemma 19, let $\hat{f}(\pi_i)$ be the estimated mean of $\pi_i$ in $\frac{16 c_M \sigma^2 \log \frac{2m}{\delta}}{\alpha^2}$ rounds,

$$\mathbb{P}\left[\left|f(\pi_i) - \hat{f}(\pi_i)\right| \geq \frac{\alpha}{4}\right] \leq \frac{\delta}{2m}.$$

For each $\pi_i$, after sampling $\frac{16 c_M \sigma^2 \log \frac{2m}{\delta}}{\alpha^2}$ times and estimate using the median of means method, plus union bound, we have with probability at least $1 - \frac{\delta}{2}$, $|f(\pi_i) - \hat{f}(\pi_i)| \leq \frac{\alpha}{4} \ \forall \pi_i$. Therefore,

in total, with probability at least $1 - \delta$, the arm $\hat{\pi}$ with the largest estimated mean has the property $\sup_{\pi^*} f(\pi^*) - \hat{f}(\hat{\pi}) \leq \frac{3}{4}\alpha$. For this arm $\hat{\pi}$, we have $\sup_{\pi^*} f(\pi^*) - f(\hat{\pi}) \leq \alpha$. ∎

Next, consider the model $M$ such that for each arm $\pi$, $M(\pi)$ is a Gaussian distribution with variance $\sigma^2$. Let $\mathcal{M}_{\mathcal{F}}^{G,\sigma^2}$ denote a set of all such models $M$ such that $f^M \in \mathcal{F}$. With the knowledge of $\mathcal{F}$, suppose the true model $M \in \mathcal{M}_{\mathcal{F}}^{G,\sigma^2}$, if there exists an algorithm that is able to identify an arm $\hat{\pi}$ such that $f^M(\hat{\pi}) \geq \sup_{\pi} f^M(\pi) - \alpha$ with probability at least $1 - \delta$ in a bounded number of rounds $\forall \alpha, \delta \in (0, 1)$ and $\forall M \in \mathcal{M}_{\mathcal{F}}^{G,\sigma^2}$, then we say function class $\mathcal{F}$ is learnable with Gaussian noise. Theorem 10 shows a lower bound for learnability of $\mathcal{F}$ with Gaussian noise.

**Theorem 10** $\gamma_{\mathcal{F},\alpha} > 0 \; \forall \alpha \in (0, 1)$ *is a necessary condition for learnability of $\mathcal{F}$ with Gaussian noise.*

**Proof** Let $\|P - Q\|_{TV}$ denote the total variation distance between distributions $P$ and $Q$. Consider $G_1$ as a Gaussian distribution with mean 0 and variance $\sigma^2$, and $G_2$ as a Gaussian distribution with mean 1 and variance $\sigma^2$. For any Gaussian distribution $G$ with a mean between 0 and 1, we define a modified distribution $G'$ such that $\|P_G - P_{G'}\|_{TV} \leq \epsilon$, where $P_G$ and $P_{G'}$ are the probability density functions of $G$ and $G'$, respectively. First, partition $P_{G'}$ into $m = w + 2$ buckets, where $w$ is a constant to be defined later in this paragraph. For the leftmost and rightmost buckets, find two points $c_1$ and $c_2$ such that $\int_{-\infty}^{c_1} P_{G_1}(x)dx = \frac{\epsilon}{4}$ and $\int_{c_2}^{\infty} P_{G_2}(x)dx = \frac{\epsilon}{4}$. Define $G'$ to be 0 in these two buckets. Consequently, $\|P_G - P_{G'}\|_{TV} \leq \frac{\epsilon}{4}$ for each of these buckets. Using Lemma 20, we know that Gaussian distributions are $L$-Lipschitz, i.e., $|P_G(x_1) - P_G(x_2)| \leq L|x_1 - x_2|$ for any Gaussian distribution $G$. For the region between $c_1$ and $c_2$, divide it into $w = \frac{L(c_2 - c_1)^2}{\epsilon}$ buckets, each with a width of $\frac{\epsilon}{L(c_2 - c_1)}$. Let $l$ and $r$ represent the left and right boundaries of a bucket. Define $G'$ in each of these middle buckets as a uniform distribution such that: $\int_l^r P_G(x)dx = \int_l^r P_{G'}(x)dx - \frac{\epsilon}{2w}$. The adjustment $-\frac{\epsilon}{2w}$ ensures that $\int_{-\infty}^{\infty} P_{G'}(x)dx = 1$. Within each middle bucket, we have $P_G(r) - P_G(l) \leq \frac{\epsilon}{c_2 - c_1}$, leading to $\|P_G - P_{G'}\|_{TV} \leq \frac{\epsilon(r-l)}{2(c_2 - c_1)} = \frac{\epsilon}{2w}$. Since there are $w$ buckets in the middle, the total variation distance in this region satisfies $\|P_G - P_{G'}\|_{TV} \leq \frac{\epsilon}{2}$. Combining the contributions from the leftmost, rightmost, and middle regions, we conclude that $\|P_G - P_{G'}\|_{TV} \leq \epsilon$ for any Gaussian distribution $G$ with a mean between 0 and 1.

Let the function $F$ represent an $n$-round algorithm $A$, where the input to $F$ is the information the algorithm receives at each round. For any $n$-round algorithm $A$, let $P$ denote the distribution of $F(G_1, \ldots, G_n, B)$ and $P'$ denote the distribution of $F(G'_1, \ldots, G'_n, B)$. Here, $G_i$ is the distribution of the original reward in the $i$-th round, $G'_i$ is the modified distribution in the $i$-th round following the modification described in the first paragraph, and $B$ represents the randomness of the algorithm. Given that $\|P_{G_i} - P_{G'_i}\|_{TV} \leq \epsilon$ for all $i$, it follows that $\|P - P'\|_{TV} \leq n\epsilon$. By setting $\epsilon \leq \frac{1}{4n}$ and $\delta \in (0, \frac{1}{2})$, the algorithm guarantees that $f^*(\hat{\pi}) \geq f^*(\pi_*) - \alpha$ under $P$ with probability greater than $\frac{1}{2}$. Consequently, under $P'$, this result holds with probability greater than $\frac{1}{4}$.

Next, we define the distribution $G''$ as follows: it first uniformly selects one of the $w + 2$ buckets and then uniformly selects a reward within the chosen bucket. Note that the conditional distribution within each bucket is identical between $G'$ and $G''$ (both are uniform distributions over a fixed region). We now demonstrate that if the learning algorithm $A$ has finite query complexity $n$, then there exists a distribution over arms such that for any function in the class, there is a non-zero probability of sampling a near-optimal arm. Specifically, let $\mathcal{M}'$ denote the model class where the

arms have reward distributions $G'$, and let $\mathcal{M}''$ denote the model class where the arms have reward distributions $G''$. We execute the algorithm $A$, but whenever $A$ pulls an arm, we respond to its queries in each round using $G''$. $\forall M' \in \mathcal{M}'$, with probability $(w+2)^{-n}$, these all agree with rewards $M'$ would respond with, and independent of which rewards $M'$ would respond with. Let $p$ be the distribution over output of $\hat{\pi}$ by this execution, we have $\forall M^* \in \mathcal{M}_{\mathcal{F}}^{G,\sigma^2}$, $\mathbb{P}_{\pi \sim p}(\sup_{\pi^*} f^{M^*}(\pi^*) - f^{M^*}(\pi) \leq \alpha) \geq \frac{1}{4}(w+2)^{-n} > 0$.

Finally, we have $\gamma_{\mathcal{F},\alpha} \geq \mathbb{P}_{\pi \sim p}(\sup_{\pi^*} f^{M^*}(\pi^*) - f^{M^*}(\pi) \leq \alpha) \geq \frac{1}{4}(w+2)^{-n} > 0$. Therefore, we have: $T \geq \Omega\left(\log \frac{1}{\gamma_{\mathcal{F},\alpha}}\right)$. ∎

**Corollary 11** *Theorem 9 and Theorem 10 together imply $\gamma_{\mathcal{F},\alpha} > 0 \,\forall \alpha \in (0,1)$ is also a characterization of learnability for $\mathcal{F}$ with unbounded noise and $\mathcal{F}$ with Gaussian noise.*

**Remark 12** *The proof of Theorem 10 for the Gaussian noise model relies on a unified discretization scheme used for the histogram approximation of distributions. As a result, this proof can be directly generalized to any noise distributions that satisfy this property. Examples include Poisson distribution, geometric distribution (after rescaling) and other most noise distributions with Lipschitz-continuous densities.*

**Corollary 13** *The following statements are equivalent:*

- *$\gamma_{\mathcal{F},\alpha} > 0 \,\forall \alpha \in (0,1)$.*

- *$\mathcal{F}$ is learnable with arbitrary noise.*

- *$\mathcal{F}$ is learnable with binary noise.*

- *$\mathcal{F}$ is learnable with Gaussian noise.*

## 5. The Variant of Decision Estimation Coefficient

### 5.1. Discussion about existing Decision Estimation Coefficient

Recent work (Foster et al., 2021, 2023) provide a well-known complexity measure for bandit learnability and even more general interactive decision making problems, which they termed as *Decision Estimation Coefficient (DEC)*. Consider the DEC formulation from Foster et al. (2023):

$$\mathbf{dec}_\varepsilon(\mathcal{M}) = \sup_{\overline{M} \in \mathrm{co}(\mathcal{M})} \inf_{p,q \in \Delta(\Pi)} \sup_{M \in \mathcal{M}} \left\{ \mathbb{E}_{\pi \sim p}\left[ f^M(\pi_M) - f^M(\pi) \right] \mid \mathbb{E}_{\pi \sim q}\left[ D_H^2(M(\pi), \overline{M}(\pi)) \right] \leq \varepsilon^2 \right\}$$

(5)

where $D_H^2$ denotes the Hellinger distance and $\pi_M := \arg\max_{\pi \in \Pi} f^M(\pi)$. For stochastic bandit problem, Foster et al. (2023) gives an upper bound for the query complexity based on $\mathbf{dec}_{\bar{\varepsilon}(T)}(\mathcal{M})$ for $\bar{\varepsilon}(T) = \tilde{\Theta}(\sqrt{\mathbf{Est}_H(T)/T})$. Here, $\mathbf{Est}_H(T)$ represents the complexity when performing online distribution estimation with the model class $\mathcal{M}$.

In this work, we are considering arbitrary noise, which is a highly complex family of distributions. It is impossible to achieve non-trivial guarantee based on distribution estimation, hence the constraint in $\mathbf{dec}_{\bar{\varepsilon}(T)}(\mathcal{M})$ does not rule out any function in the function class $\mathcal{F}$. Given this fact, consider function class $\mathcal{F}_1$, the set of all functions on two arms. It is easy to show $\mathbf{dec}_{\bar{\varepsilon}(T)}(\mathcal{M}_{\mathcal{F}_1}) = \frac{1}{2}$

since at best distribution $p$ in Eq.(5) is uniform on two arms. $\mathcal{F}_1$ is learnable since the arm number is finite. As a contrast, consider function class $\mathcal{F}_2 = \{\frac{1}{2}\mathbb{I}_z, z \in \mathbb{R}\}$. Note that for any distribution $p$, there exists some arm $\pi$ where $p$ has 0 probability mass, and there is a model $M$ that has $f^M(\pi) = \frac{1}{2}$. Thus we have $\mathbf{dec}_{\bar{\varepsilon}(T)}(\mathcal{M}_{\mathcal{F}_2}) = \frac{1}{2}$. $\mathcal{F}_2$ is not learnable. There exists a learnable function class $\mathcal{F}_1$ and a non-learnable class $\mathcal{F}_2$ that $\mathbf{dec}_{\bar{\varepsilon}(T)}(\mathcal{M})$ have the same value. This indicates it cannot characterize learnability in stochastic bandit with arbitrary noise. Even with specific (such as binary or Gaussian) noise, the complexity of online distribution estimation can still be large. In these cases, $\mathbf{dec}_{\bar{\varepsilon}(T)}(\mathcal{M})$ is not made trivially vacuous by the richness of the noise model. However, it is still unclear whether it could characterize the learnability of stochastic bandit.

## 5.2. A new variant of Decision Estimation Coefficient

In this section, inspired by our proposed *generalized maximin volume*, we give a new variant of Decision Estimation Coefficient that can also characterize the learnability of stochastic noisy bandits. Our modification involves two key changes: first, we replace the Hellinger distance between models to square loss between functions. Since our focus is on arbitrary noise setting, we replace the online distribution estimation oracle to the online regression oracle, and update its corresponding guarantees accordingly. Second, we change the expectation $\mathbb{E}_{\pi \sim p}[\sup_{\pi^*} f(\pi^*) - f(\pi)]$ to probability $\mathbb{P}_{\pi \sim p}(\sup_{\pi^*} f(\pi^*) - f(\pi) > \alpha)$ in Eq.(5). This change is essential for ensuring learnability, similar to the role of *generalized maximin volume*.

Our variant of the Decision Estimation Coefficient is inspired by Theorem 8, which demonstrates that certain function classes allow adaptive algorithms to achieve better query complexity than non-adaptive algorithms. This insight motivates us to propose a generic adaptive algorithm and integrate this information into *generalized maximin volume*. While the algorithm is not always optimal, it narrows the gap between upper and lower bounds for many function classes. This improvement is reflected in our analysis using the proposed variant of DEC. Specifically, when the function class $\mathcal{F}$ admits efficient online regression, the constraint in DEC quickly diminishes to a small value over a short time horizon $T$. As a result, DEC effectively becomes a localized version of the *generalized maximin volume*.

Formally, we introduce our proposed variant of DEC: given $\alpha \in (0,1)$, let

$$\mathbf{dec}_{\varepsilon,\alpha}(\mathcal{F},\overline{f}) = \inf_{p,q \in \Delta(\Pi)} \sup_{f \in \mathcal{F}} \left\{ \mathbb{P}_{\pi \sim p}\left(\sup_{\pi^*} f(\pi^*) - f(\pi) > \alpha\right) \middle| \mathbb{E}_{\pi \sim q}\left[(f(\pi) - \overline{f}(\pi))^2\right] \leq \varepsilon^2 \right\}$$

Further, let $\mathrm{co}(\mathcal{F})$ denote the convex hull of function class $\mathcal{F}$, define

$$\mathbf{dec}_{\varepsilon,\alpha}(\mathcal{F}) = \sup_{\overline{f} \in \mathrm{co}(\mathcal{F})} \mathbf{dec}_{\varepsilon,\alpha}(\mathcal{F},\overline{f})$$

**Definition 14 (Online regression oracle for $\mathcal{F}$)** *At each time $t \in [T]$, an online regression oracle* **Reg** *for $\mathcal{F}$ returns, given*

$$\mathfrak{H}^{t-1} = (\pi^1, r^1), ..., (\pi^{t-1}, r^{t-1})$$

*with $r^i \sim M^*(\pi^i)$ and $\pi^i \sim p^i$, an estimator $\widehat{f}^t \in \mathrm{co}(\mathcal{F})$ such that whenever $f^* \in \mathcal{F}$ (Equivalently, $M^* \in \mathcal{M}_{\mathcal{F}}$),*

$$\mathbf{EST}(T) := \sum_{t=1}^{T} \mathbb{E}_{\pi^t \sim p^t}\left[(f^*(\pi^t) - \widehat{f}^t(\pi^t))^2\right] \leq \mathbf{EST}(T, \delta) \tag{6}$$

*with probability at least* $1 - \delta$, *where* $\boldsymbol{EST}(T, \delta)$ *is a known upper bound.*

Next, we will describe another characterization of learnability for stochastic bandits with arbitrary noise based on our variant of the Decision-Estimation-Coefficient. Let $T \in \mathbb{N}$, define $\overline{\textbf{EST}} := \textbf{EST}\left(\frac{2T}{\lceil \log 4/\delta \rceil}, \frac{\delta}{4 \lceil \log 4/\delta \rceil}\right)$ and set $\overline{\varepsilon}(T) := 8\sqrt{\frac{\lceil \log 4/\delta \rceil}{T} \overline{\textbf{EST}}}$. Then we have:

**Theorem 15** $\mathcal{F}$ *is learnable with arbitrary noise if and only if* $\forall \alpha \in (0,1), \exists T \in \mathbb{N}, \boldsymbol{dec}_{\overline{\varepsilon}(T),\alpha}(\mathcal{F}) < 1$.

---

**Algorithm 3** Learning algorithm based on DEC Variant

---

**Input:** A number $T \in \mathbb{N}$.
Failure probability $\delta > 0$.
Online regression oracle **Reg**.
Define $L := \lceil \log 4/\delta \rceil$, $J := \frac{T}{L+1}$, and $\overline{\textbf{EST}} := \textbf{EST}\left(\frac{2T}{\lceil \log 4/\delta \rceil}, \frac{\delta}{4\lceil \log 4/\delta \rceil}\right)$.
  Let $\mathcal{H}_{p,\varepsilon}(\bar{f}) := \{f \in \mathcal{F} | \mathbb{E}_{\pi \in p}[(f(\pi) - \bar{f}(\pi))^2] \leq \varepsilon^2\}$.
  Set $\overline{\varepsilon}(T) := 8\sqrt{\frac{\lceil \log 4/\delta \rceil}{T} \overline{\textbf{EST}}}$.
/*exploration phase
  **for** $t = 1, 2, \cdots, J$ **do**
    Obtain estimate $\widehat{f}^t = \textbf{Reg}\left(\{(\pi^i, r^i)\}_{i=1}^{t-1}\right)$.
      Compute

$$(p^t, q^t) = \arg\min_{p,q \in \Delta(\Pi)} \sup_{f \in \mathcal{H}_{q,\overline{\varepsilon}(T)}(\widehat{f}^t)} \mathbb{P}_{\pi \sim p}\left(\sup_{\pi^*} f(\pi^*) - f(\pi) > \frac{\alpha}{2}\right).$$

      with the convention that the value is zero if $\mathcal{H}_{q,\overline{\varepsilon}(T)}(\widehat{f}^t) = \emptyset$.
      Sample decision $\pi^t \sim q^t$ and update regression oracle **Reg** with $(\pi^t, r^t)$.
  **end**
/* exploitation phase
  Sample $L$ indices $t_1, ..., t_L \sim \text{Unif}([J])$ independently.
  For each $\ell \in [L]$, draw $J$ independent samples $\pi^1_\ell, ..., \pi^J_\ell \sim q^{t_\ell}$, and observe $(\pi^j_\ell, r^j_\ell)$ for each $j \in [J]$.
  For each $\ell \in [L]$ and $j \in [J]$, compute

$$\widetilde{f}^j_\ell := \textbf{Reg}(\{(\pi^i_\ell, r^i_\ell)\}_{i=1}^{j-1}),$$

and let $\widetilde{f}_\ell := \frac{1}{J} \sum_{j=1}^{J} \widetilde{f}^j_\ell$.
Set $\hat{p} := p^{t_{\hat\ell}}$, where $\hat{\ell} := \arg\min_{\ell \in [L]} \mathbb{E}_{\pi \sim q^{t_\ell}}[(\widehat{f}^{t_\ell}(\pi) - \widetilde{f}_\ell(\pi))^2]$.
Let $\gamma = 1 - \sup_{f \in \mathcal{H}_{q,\overline{\varepsilon}(T)}(\widehat{f}^{t_{\hat\ell}})} \mathbb{P}_{\pi \sim \hat{p}}(\sup_{\pi^*} f(\pi^*) - f(\pi) > \frac{\alpha}{2})$.
Sample $m = \frac{1}{\gamma} \log \frac{2}{\delta}$ arms from $\hat{p}$: $\pi_1, \pi_2, ..., \pi_m$.
For each $\pi_i$, query $\frac{8}{\alpha^2} \log \frac{4m}{\delta}$ times. Let $\hat{f}(\pi_i)$ denote the empirical mean of arm $\pi_i$ in those rounds.
**Output:** The arm $\hat{\pi}$ with largest empirical mean $\hat{f}(\pi_i)$.

---

**Theorem 16 (Upper Bound)**

$$\forall \alpha \in (0, 1), \exists T \in \mathbb{N}, \boldsymbol{dec}_{\overline{\varepsilon}(T),\alpha}(\mathcal{F}) < 1$$

is a sufficient condition for learnability of function class $\mathcal{F}$ with arbitrary noise. In addition,
$QC(\mathcal{F}, \alpha, \delta) = O\left(T + \frac{8}{(1 - \boldsymbol{dec}_{\overline{\varepsilon}(T), \alpha/2}(\mathcal{F}))\alpha^2} \log \frac{2}{\delta} \log \left(\frac{4}{\delta(1 - \boldsymbol{dec}_{\overline{\varepsilon}(T), \alpha/2}(\mathcal{F}))} \log \frac{2}{\delta}\right)\right).$

**Proof** We begin by analyzing the exploitation phase. Recall that we set $J := \frac{T}{\lceil \log 4/\delta \rceil + 1} \geq \frac{T}{2L}$. We have that with probability at least $1 - \frac{\delta}{4L}$,

$$\sum_{t=1}^{J} \mathbb{E}_{\pi^t \sim q^t}\left[\left(f^*(\pi^t) - \widehat{f}^t(\pi^t)\right)^2\right] \leq \mathbf{EST}\left(J, \frac{\delta}{4L}\right) \leq \overline{\mathbf{EST}}.$$

We denote this event by $\mathscr{E}_0$, and condition on it going forward. Since we have $\overline{\varepsilon}(T)^2 \geq \frac{32}{J}\overline{\mathbf{EST}}$ by definition, it follows from Markov's inequality that if $s \in [J]$ is chosen uniformly at random, then with probability at least $1/2$,

$$\mathbb{E}_{\pi^s \sim q^s}\left[\left(f^*(\pi^s) - \widehat{f}^t(\pi^s)\right)^2\right] \leq \frac{\overline{\varepsilon}(T)^2}{16}. \tag{7}$$

Going forward, our aim is to show that the exploitation phase identifies such an index $s \in [J]$. Indeed, for any $s \in [J]$ such that the inequality (7) holds, we have $f^* \in \mathcal{H}_{q^s, \overline{\varepsilon}(T)}(\widehat{f}^s)$, and hence

$$\mathbb{P}_{\pi \sim p^s}\left(\sup_{\pi^*} f^*(\pi^*) - \widehat{f}^s(\pi) > \frac{\alpha}{2}\right) \leq \boldsymbol{dec}_{\overline{\varepsilon}(T), \alpha/2}(\mathcal{F}, \widehat{f}^s) \leq \boldsymbol{dec}_{\overline{\varepsilon}(T), \alpha/2}(\mathcal{F})$$

To proceed, first observe that for the uniformly sampled indices $t_1, \ldots, t_L \in [J]$, a standard confidence boosting argument implies that with probability at least $1 - 2^{-L} \geq 1 - \frac{\delta}{4}$, there is some $\ell \in [L]$ so that (7) is satisfied with $s = t_\ell$. We denote this event by $\mathscr{F}$.

Next, recall the definition $\widetilde{f}_\ell = \frac{1}{J}\sum_{j=1}^{J} \widetilde{f}_\ell^j$. We have that for each $\ell \in [L]$, there is an event that occurs with probability at least $1 - \frac{\delta}{4L}$, denoted by $\mathscr{E}_\ell$, such that under $\mathscr{E}_\ell$ we have

$$
\begin{aligned}
\mathbb{E}_{\pi \sim q^{t_\ell}}\left[(f^*(\pi) - \widetilde{f}_\ell(\pi))^2\right] &= \mathbb{E}_{\pi \sim q^{t_\ell}}\left[\left(f^*(\pi) - \frac{1}{J}\sum_{t=1}^{J} \widetilde{f}_\ell^t(\pi)\right)^2\right] \\
&\leq \frac{1}{J}\sum_{t=1}^{J} \mathbb{E}_{\pi \sim q^{t_\ell}}\left[\left(f^*(\pi) - \widetilde{f}_\ell^t(\pi)\right)^2\right] \\
&\leq \frac{\mathbf{EST}(J, \delta/4L)}{J} \\
&\leq \frac{\overline{\mathbf{EST}}}{J} \\
&\leq \frac{\overline{\varepsilon}(T)^2}{32}
\end{aligned}
\tag{8}
$$

We define $\mathscr{E} := \mathscr{F} \cap \bigcap_{\ell=0}^{L} \mathscr{E}_\ell$, so that $\mathscr{E}$ occurs with probability at least $1 - \frac{(L+1)\delta}{4L} - \frac{\delta}{4} \geq 1 - \frac{\delta}{2}$.

We now show that the exploitation phase succeeds whenever the event $\mathscr{E}$ holds. By the triangle inequality for square loss, letting $\ell \in [L]$ be any index such that (7) is satisfied with $s = t_\ell$, we have

$$\mathbb{E}_{\pi \sim q^{t_\ell}}\left[\left(\widetilde{f}_\ell(\pi) - \widehat{f}^{t_\ell}(\pi)\right)^2\right] \leq 2\left(\mathbb{E}_{\pi \sim q^{t_\ell}}\left[\left(f^*(\pi) - \widetilde{f}_\ell(\pi)\right)^2 + \left(f^*(\pi) - \widehat{f}^{t_\ell}(\pi)\right)^2\right]\right) \leq \frac{\overline{\varepsilon}(T)^2}{4} \tag{9}$$

In addition, based on the definition of $\hat{\ell}$,

$$\mathbb{E}_{\pi \sim q^{t_{\hat{\ell}}}}\left[\left(\widetilde{f}_{\hat{\ell}}(\pi) - \widehat{f}^{t_{\ell}}(\pi)\right)^2\right] \leq \mathbb{E}_{\pi \sim q^{t_{\ell}}}\left[\left(\widetilde{f}_{\ell}(\pi) - \widehat{f}^{t_{\ell}}(\pi)\right)^2\right] \leq \frac{\overline{\varepsilon}(T)^2}{4} \tag{10}$$

Using triangle inequality again, we obtain under the event $\mathscr{E}$,

$$\mathbb{E}_{\pi \sim q^{t_{\hat{\ell}}}}\left[\left(f^*(\pi) - \widehat{f}^{t_{\hat{\ell}}}(\pi)\right)^2\right] \leq 2\left(\frac{\overline{\varepsilon}(T)^2}{4} + \frac{\overline{\varepsilon}(T)^2}{32}\right) \leq \overline{\varepsilon}(T)^2 \tag{11}$$

This means that $f^* \in \mathcal{H}_{q^{t_{\hat{\ell}}}, \overline{\varepsilon}(T)}(\widehat{f}^{t_{\hat{\ell}}})$. In addition,

$$\sup_{f \in \mathcal{H}_{q^{t_{\hat{\ell}}}, \overline{\varepsilon}(T)}(\widehat{f}^{t_{\hat{\ell}}})} \mathbb{P}_{\pi \sim p^{t_{\hat{\ell}}}}\left(f(\pi_M) - f(\pi) > \frac{\alpha}{2}\right) = \inf_{p,q \in \Delta(\Pi)} \sup_{f \in \mathcal{H}_{q, \overline{\varepsilon}(T)}(\widehat{f}^{t_{\hat{\ell}}})} \mathbb{P}_{\pi \sim p}\left(f(\pi_M) - f(\pi) > \frac{\alpha}{2}\right)$$

$$= \mathbf{dec}_{\overline{\varepsilon}(T), \frac{\alpha}{2}}(\mathcal{F}, \widehat{f}^{t_{\hat{\ell}}})$$

$$\leq \sup_{\overline{f} \in \mathrm{co}(\mathcal{F})} \mathbf{dec}_{\overline{\varepsilon}(T), \frac{\alpha}{2}}(\mathcal{F}, \overline{f})$$

$$= \mathbf{dec}_{\overline{\varepsilon}(T), \frac{\alpha}{2}}(\mathcal{F})$$

$$\tag{12}$$

Therefore, with probability $1 - \frac{\delta}{2}$, we have $f^*$ is contained in $\mathcal{H}_{q^{t_{\hat{\ell}}}, \overline{\varepsilon}(T)}(\widehat{f}^{t_{\hat{\ell}}})$ and $\gamma \geq 1 - \mathbf{dec}_{\overline{\varepsilon}(T), \frac{\alpha}{2}}(\mathcal{F}) > 0$. When we sample $\pi$ from distribution $\hat{p}$ $m = \frac{1}{\gamma}\log(\frac{2}{\delta})$ times, we have:

$$\mathbb{P}\left(\exists \pi_i : \sup_{\pi^*} f^*(\pi^*) - f^*(\pi_i) \leq \alpha/2\right)$$

$$= 1 - \mathbb{P}\left(\forall \pi_i : \sup_{\pi^*} f^*(\pi^*) - f^*(\pi_i) > \alpha/2\right)$$

$$\geq 1 - (1 - \gamma)^{\frac{1}{\gamma}\log(\frac{2}{\delta})} \tag{13}$$

$$\geq 1 - e^{-\log(\frac{2}{\delta})}$$

$$= 1 - \frac{\delta}{2}$$

Therefore, with probability at least $1 - \frac{\delta}{2}$, $\exists \pi_i, \sup_{\pi^*} f^*(\pi^*) - f^*(\pi_i) \leq \frac{\alpha}{2}$. Then, based on Lemma 18, let $\hat{f}(\pi_i)$ be the empirical mean of $\pi_i$ in $\frac{8}{\alpha^2}\log\frac{4m}{\delta}$ rounds,

$$\mathbb{P}\left[\left|f^*(\pi_i) - \hat{f}^*(\pi_i)\right| \geq \frac{\alpha}{4}\right] \leq 2e^{-2\frac{8}{\alpha^2}\log\frac{4m}{\delta}\frac{\alpha^2}{16}} = \frac{\delta}{2m}$$

Then, based on union bound, with probability $1 - \frac{\delta}{2}$, for all arms $\pi_i$, its empirical mean is within $\frac{\alpha}{4}$ from their true mean. Then in total, with probability $1 - \delta$, the arm $\hat{\pi}$ with largest estimated mean has the property $\sup_{\pi^*} f(\pi^*) - \hat{f}(\hat{\pi}) \leq \frac{3}{4}\alpha$. Therefore, for arm $\hat{\pi}$, $\sup_{\pi^*} f(\pi^*) - f(\hat{\pi}) \leq \alpha$ as desired.

∎

**Theorem 17 (Lower Bound)**

$$\forall \alpha \in (0,1), \exists\, T \in \mathbb{N}, \boldsymbol{dec}_{\overline{\varepsilon}(T),\alpha}(\mathcal{F}) < 1$$

*is a necessary condition for learnability of function class $\mathcal{F}$ with arbitrary noise.*

**Proof** For any learnable function class $\mathcal{F}$, we have $\sup_{p\in\Delta(\Pi)} \inf_{f\in\mathcal{F}} \mathbb{P}_{\pi\sim p}(\sup_{\pi^*} f(\pi^*) - f(\pi) \leq \alpha) > 0 \quad \forall \alpha$ by Theorem 3. Equivalently, we have $\inf_{p\in\Delta(\Pi)} \sup_{f\in\mathcal{F}} \mathbb{P}_{\pi\sim p}(\sup_{\pi^*} f(\pi^*) - f(\pi) > \alpha) < 1 \quad \forall \alpha$. Following this, we have: $\sup_{\overline{f}\in\mathrm{co}(\mathcal{F})} \inf_{p,q\in\Delta(\Pi)} \sup_{f\in\mathcal{F}} \{\mathbb{P}_{\pi\sim p}(\sup_{\pi^*} f(\pi^*) - f(\pi) > \alpha) | \mathbb{E}_{\pi\sim q}[(f(\pi) - \overline{f}(\pi))^2] \leq \varepsilon^2\} < 1 \,\forall\alpha\forall\varepsilon$, since any constraint would only make the maximum value smaller, which will be still smaller than 1. ∎

At last, we wrap up this section by giving an illustrating example to show how our proposed DEC could improve query complexity in some cases compared to *generalized maximin valume*.

**Example 4** *Consider the function class $\mathcal{F}$ constructed in Theorem 7 and set $N = 1$. For clarity, we denote $\gamma_{\mathcal{F},\alpha}$ as $\gamma$ and $\boldsymbol{dec}_{\overline{\varepsilon}(T),\alpha}(\mathcal{F})$ as $\boldsymbol{dec}$. We claim that the upper bound on $\boldsymbol{dec}$ in Theorem 16 achieves a near-optimal query complexity of $\mathrm{polylog}\left(\frac{1}{\gamma}\right)$.*

*First, for any function $f \in \mathcal{F}$, there are $\log\left(\frac{1}{\gamma}\right)$ non-zero mean values, which are $\frac{1}{3}$, $\frac{2}{3}$ or 1. Observing at most $O\left(\log\left(\frac{\log(\frac{1}{\gamma})}{\delta}\right)\right)$ samples for each value suffices to deduce the correct outcome. This implies that $\boldsymbol{EST}(T,\delta) = \mathrm{polylog}(\frac{1}{\gamma})$. Consequently, $\overline{\varepsilon}(T) = \tilde{O}\left(\sqrt{\frac{\log(\frac{1}{\gamma})}{T}}\right)$.*

*Next, we bound $\boldsymbol{dec}$ for a sufficiently small $\varepsilon = \frac{1}{10}$. If $\overline{f} \in \mathcal{F}$, assigning $p$ as a single point mass on the arm with reward 1 ensures $\boldsymbol{dec} = 0$. For $\overline{f} \in \mathrm{co}(\mathcal{F}) \setminus \mathcal{F}$, we define $\boldsymbol{dec}$ for any fixed $\overline{f}$ as 0 if there exists $q$ such that the constraint is infeasible for all $f \in \mathcal{F}$. Otherwise, if every $q$ has a non-empty set of functions in $\mathcal{F}$ $\varepsilon$-close to $\overline{f}$ under $q$, then $\overline{f}$ is very close to some $f \in \mathcal{F}$. In this case, there exists a leaf with near 1 value where $p$ can place a single point mass, ensuring $\boldsymbol{dec} = 0$.*

*To justify this claim for $\overline{f} \in \mathrm{co}(\mathcal{F}) \setminus \mathcal{F}$, note that at the root level, if $\overline{f}$ is not close to $\frac{1}{3}$ or $\frac{2}{3}$, then a distribution $q$ with a single point mass at the root makes the constraint set infeasible. Therefore, $\overline{f}$ must be close to $\frac{1}{3}$ or $\frac{2}{3}$ at the root. Without loss of generality, assume $\overline{f} = \frac{1}{3}$ at the root. If $\overline{f}$ is not close to $\frac{1}{3}$ or $\frac{2}{3}$ at the left child, then $q$ with point masses at the root and left child will make the constraint infeasible. Proceeding recursively, $\overline{f}$ must follow the appropriate $\frac{1}{3}$ or $\frac{2}{3}$ values along some branch in the tree (close to $\frac{1}{3}$ when going left, $\frac{2}{3}$ when going right). If $\overline{f}$ is not close to 1 at the corresponding leaf, then point masses at the leaf's parent and the leaf render the constraint set infeasible. Otherwise, $\overline{f}$ is close to 1 at some leaf, in which case a single point mass at that leaf ensures the constraint set contains only the function corresponding to the leaf, yielding $\boldsymbol{dec} = 0$.*

## 6. Conclusions and Future Directions

In this work, we have given a complete characterization of learnability for stochastic noisy bandits and further explored the range of possible values of the optimal query complexity. One interesting direction for future work could be extending these results to other variants of learning problem with bandit feedback, such as contextual bandits (Langford and Zhang, 2007; Li et al., 2010; Slivkins, 2011a; Agarwal et al., 2014; Agrawal and Devanur, 2016; Krishnamurthy et al., 2020; Foster and

Rakhlin, 2020; Foster and Krishnamurthy, 2021), dueling bandits (Yue and Joachims, 2009), combinatorial bandits (Cesa-Bianchi and Lugosi, 2012; Chen et al., 2013; Combes et al., 2015; Chen et al., 2016).

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

## Appendix A. Auxiliary Lemma

**Lemma 18** *Let $X_1, ..., X_n$ be independent random variable such that $0 \leq X_i \leq 1$ almost surely. Let $\bar{Z} = \frac{X_1 + ... + X_n}{n}$. Then:*

$$\mathbb{P}(|\bar{Z} - \mathbb{E}[\bar{Z}]| \geq t) \leq 2e^{-2nt^2}$$

*Equivalently, $\forall \delta \in (0, 1)$, with probability at least $1 - \delta$,*

$$|\bar{Z} - \mathbb{E}[\bar{Z}]| \leq \sqrt{\frac{1}{2n} \ln \left( \frac{2}{\delta} \right)}.$$

**Lemma 19** *Assume that the sample size $n = KB$, where $K$ is the number of subsamples and $B$ is the size of each subsample. We first randomly split the data into $K$ subsample and compute the mean using each subsample, which leads to a set of estimators. Each estimator is based on $B$ observations. The median-of-means estimator is the median of all these estimators. For any distribution $D$ with mean $\mu$ and standard deviation $\sigma$, the median-of-means estimate $\kappa$, on input $n$ samples, satisfies*

$$\mathbb{P}\left( |\kappa - \mu| > c_M \sigma \sqrt{\frac{\log \frac{1}{\delta}}{n}} \right) \leq \delta,$$

*where $c_M$ is an absolute constant.*

**Lemma 20 (Fact)** *Gaussian distribution is $L$-lipschitz, where $L$ is a constant depending on variance $\sigma^2$.*

