# OpenReview forum: "A Complete Characterization of Learnability for Stochastic Noisy Bandits"
_algorithmiclearningtheory.org/ALT/2025/Conference — ALT 2025_

### Official Review · Reviewer_ZqVj · 2024-10-26

**Rating:** 7
**Confidence:** 3

**Review:**

This paper studies the question of learnability of stochastic bandits for the PAC learning framework. The noiseless setting was studied in earlier work [Hanneke and Yang 2023] and showed that learnability can be undecidable in that case. Formally, a function class $\mathcal F$ from arms to rewards is learnable (for the noiseless case) if for any tolerance, there is a finite horizon $T$ and some algorithm that finds a near-optimal arm with high probability using $T$ arm queries, for any reward function within the function class $\mathcal F$.
In the main stochastic case they consider, the algorithm should find a near-optimal arm (up to $\alpha$) for any stochastic rewards within $[0,1]$ such that their mean reward function is within the function class $\mathcal F$ (realizable). They provide a characterization of learnability for stochastic bandits in terms of quantities $\gamma_{\mathcal F,\alpha}$ and characterize the range of possible optimal rates depending on this quantity (basically between $\log 1/\gamma_{\mathcal F,\alpha}$ and $1/\gamma_{\mathcal F,\alpha}$. They also provide an alternative characterization in terms of a variant of DEC.
Last, they show that this quantity also characterizes learnability when rewards may be unbounded but are supposed to have a bounded variance.

The question is definitely of interest and closes an open question from [Hanneke and Yang 2023]. In particular, while in the noiseless learnability is undecidable, for the noisy case there is a very simple characterization, which (almost disappointingly) essentially states that a good algorithm is the following: given a good prior on arms (existence of which is guaranteed by the characterization), simply test random samples of arms according to this prior and output the best arm observed.
The writing of the paper (english and clarity) could be largely improved, however, specifically starting from section 3. The ideas are quite simple but this made reading some of the proofs impossible. See below for some more detailed comments/questions. I am also confused about the utility of Section 5 which gives a DEC-variant as characterization. I understand the goal of relating the results to the DEC literature but I am not sure which insights this variant provide on the problem compared to the first characterization. It is a lot more complex, requires a significant amount notations and definitions (their presentation is also not the clearest), and hides the simple structure that was revealed by the first characterization (that there is a single good prior on arms).

Minor and detailed comments:

- p4 second paragraph of the proof of Thm 3, sentence starting with "We construct a distribution..." has grammatical issues.

- proof of Thm 4. The writing is quite confusing. From what is written, I imagine that we have fixed $\gamma\in\{1/i,i\in \mathbb N\}$ and I follow the proof replacing all occurrences of $\gamma_{\mathcal F,\alpha}$ with this $\alpha$ (it does not make sense to define the tree in terms of $\gamma_{\mathcal F,\alpha}$ while we didn't even define $\mathcal F$ yet). The actual link with the quantity $\gamma_{\mathcal F,\alpha}$ is not given within the proof. I imagine the argument is simply that the optimal prior $p$ on arms is the uniform on all arms in the leaf buckets, which is $N/\gamma$. In that case, I get $\gamma_{\mathcal F,\alpha} = \gamma/N$. This changes the parametrization of the example a little bit (although the final result is the same).

- p6 "The query of the adaptive" -> "The query of an adaptive"? Similarly for the whole paragraph.

- p6 Proof of Thm5 "for non-adptive algorithm" -> add "s"

- p7 Proof of Thm 5, what is the point of the extra $1/10$ in the last paragraph of the proof? Shouldn't this be $1-2/5$ the complementary event of the one on which $N\geq 2$?

- The whole proof of Thm 7 is very hard to read. In addition to grammatical mistakes, a few notations are not defined when used. What is $F$? What is $n$? Is it not the number of rounds $T$? What is $G'$ in the last paragraph?

- Why do we need to use Thm1 in Corollary 8 if the statement is about the unbounded reward case?

- Algorithm 3: line in the middle of the exploitation phase. When defining $\gamma$, this uses $f^\star$, but we do not have access to it, correct? Next line, what is $p$? Is it $\hat p?$

- p9 proof of Thm11. What is $\mathcal H$?

**Paper Award:**

No

---

> ### Author Response · Authors · 2024-11-21
>
> Thank you for your detailed feedback!
>
> For the dec part, our modification has two parts: 1. We change distribution estimation oracle to a regression oracle; since our focus is function classes with arbitrary zero-mean noise, distribution estimation would not make sense anymore; 2. We change the expectation part $E_{\pi \sim p}[f(\pi^*)-f(\pi)]$ to probability $P_{\pi \sim p}(f(\pi^*)-f(\pi)>\alpha)$. The second part is crucial for guaranteeing that it characterizes learnability (see response to reviewer uSop), as in our $\gamma$ complexity measure.
>
> For the second bullet, there is a typo in the first paragraph. We need to choose $\alpha \in (0,1/3)$. Let $K$ be the number of leaves (denote optimal arms). Since all optimal arms have value 1, choose $p$ in $\gamma$ to be a uniform distribution over optimal arms (leaves), then  $\gamma=1/K$. $K=1/\gamma$. This gives the correct construction.
>
> For the fifth bullet, let $N$ denote the size of the version space. $E[\frac{1}{N}]=E[\frac{1}{N}|N\geq 2]P(N\geq 2)+E[\frac{1}{N}|N=1]P(N=1)$. Since $n=1/(10\gamma)$, $1/10$ comes from $N=1$ case, which is the probability it hits the target function within $n$ rounds. This probability is $\leq 1/(10\gamma)/(1/\gamma)=\frac{1}{10}$.  Also, it should be with probability at least $2/(5\gamma)/(1/(2\gamma))=4/5$, $N\geq 2$.  So  $E[\frac{1}{N}]\leq 1\cdot 1/10+4/5 \cdot 1/2=  1/2$.  The final probability is $ \geq 1-1/2=1/2.$
>
> For the sixth bullet, $F$ is a function that we use to denote the algorithm, and the input of this function is the information the algorithm receives each round. So we can argue it using TV distance that if the input does not change a lot, the output of the algorithm (function $F$) does not change so much. $n$ and $T$ denote the same meaning. $G'$ is the modified distribution we constructed in the first paragraph.
>
> For the seventh bullet, we do not actually need theorem 1 in corollary 8.
>
> For the eighth bullet, we do not have access to $f^*$. We should modify
> $\gamma=1- \sup_{f \in H(\hat{f})} P_{\pi \sim \hat{p}}(f(\pi_M)-f(\pi) > \alpha/2)$.
> $p$ should be $\hat{p}$. Thanks for the correction.
>
> For the last bullet,
> $H_{p,\varepsilon}(\bar{f})=${$ f \in \mathcal{F}|E_{\pi \sim p}[(f(\pi)-\bar{f}(\pi))^2]\leq \varepsilon^2 $}.

---

> > ### Comment · Reviewer_ZqVj · 2024-11-28
> >
> > Thank you for your responses. I hope the necessary changes to the proofs can be made for future revisions (in addition to a very careful pass for correctness/english).
> >
> > I appreciate your comments for the DEC-like characterization. My question was not really about why is it different from DEC, but rather why is this section important/necessary? I understand that comparing it to the DEC can be useful, but the new format of the characterization is not readable compared to the previous characterization and "misses the point/intuition" that was made in the previous one. I would rather view this as a comment (the characterization generalizes the previous DEC results) rather than a useful statement. I may have missed something about this, so please let me know if this is the case.

---

> > > ### Author Response · Authors · 2024-11-30
> > >
> > > Thank you for your feedback. Regarding our motivation of including the DEC variant part, although this section is not part of the core message of the paper, the intention is to attempt to reduce the exponential gap in the query complexity for some classes. Our work shows that there are function classes where adaptive algorithms can achieve better query complexity than non-adaptive algorithms (Theorem 5), and this motivates us to propose a generic adaptive algorithm. We do not claim this algorithm is always optimal. However, it reduces the gap for some classes, and this is reflected in our analysis based on our proposed variant of the DEC: When the function class admits easy online regression, the epsilon constraint in the DEC will decrease to a small value in a short time $T$; the DEC then effectively becomes a localized version of our gamma quantity.
> > >
> > > In particular, we note that for the function class $\mathcal{F}$ from Theorem 5 (showing advantages of adaptivity), our Theorem 11 (query complexity based on our variant of the DEC) admits a query complexity $polylog(1/\gamma)$ (ignoring $polylog(1/\delta)$ here and below for simplicity). This follows from noting $\textbf{EST}(T,\delta) = polylog(1/\gamma)$ for this class (for any $f$ in $\mathcal{F}$, there are only $\log(1/\gamma)$ non-zero mean values, and these values are each either $1/3$, $2/3$, or $1$, hence we can only observe at most $O(\log(\log(1/\gamma)/\delta))$ samples of each before we can deduce their correct value); moreover we can argue that (for this class $\mathcal{F}$) for $\varepsilon < 1/10$ (say), for any $\bar{f}$ in $co(\mathcal{F})$ for which the constraint in DEC is feasible for all $q$ (which we require of $\bar{f}$), $\bar{f}$ must be very close to some function $f$ in $\mathcal{F}$; thus, letting $p$ be a single point mass on the leaf corresponding to this $f$, the DEC value is 0.  We will include the details of this example, and additional examples of function classes illustrating such advantages, in the final version of the paper.

---

### Official Review · Reviewer_VKQB · 2024-11-02
**Arguably a fundamental result, but significance is limited.**

**Rating:** 6
**Confidence:** 5

**Review:**

The paper is on "learnability" applied to "structured" stochastic bandits. Here, the algorithm is given a function class $F$: a class of possible functions from arms to expected rewards; arbitrary reward distributions are allowed. $F$ is called "learnable" if, essentially, one can PAC-learn the best arm with bounded query complexity: i.e., if one can learn an alpha-optimal arm with probability at least 1-delta in some bounded time, for any given alpha, delta>0.

The main result is a simple "if and only if" characterization of learnability (via some sup-inf expression). The basic version applies to bounded rewards (and in fact holds when restricted to binary rewards). This is extended to unbounded rewards with reward distributions of fixed variance, and also holds when restricted to Gaussian rewards.

Interestingly, prior work (Hanneke and Yang (2023)) finds that learnability plays out very differently when rewards are deterministic: it is undecidable (within ZFC), let alone lacks a simple characterization.

Extension #1: the optimal query complexity $Q^*$ is crudely characterized: placed into an interval $[T, exp(T)]$, for some known T.

Extension #2: an alternative characterization is derived when the function class admits a regression oracle. This condition involves a variant of DEC "Decision-Estimation Coefficient from prior work of Foster et al (2021, 2022)).

MAJOR COMMENTS

[+] Learnability is arguably a fundamental question in structured bandits. A simple characterization is a very "cool" conceptual contribution, and should be published.

[-] Practical significance is limited because learnability is already well-understood for any specific function class anyone cared to study earlier.

[-] The significance of the extensions is unclear: #1 is extremely crude, and the expression in #2 seems quite a bit more complex, even if DEC-like.

The characterization's significance could be really enhanced via examples: how learnability plays out for some specific (interesting) function classes.  Ideally, also how resolving these examples via the characterization is much simpler than resolving them "from first principles". Absent such examples, the significance is less clear.

The proofs are not complicated (which is both a "pro" and a "con"). The main "lower bound" proof does something clever (even though the proof itself is short), whereas the main "upper bound" seems quite "easy". The other results are more technical, but do not seem to involve new ideas.

"Non-learnability" is not the end of the story, as one can usually address it via "smoothing", as per [KLSZ20]. That is: consider "smoothed arms" (i.e., perturbed an arm by a small noise), and compete with "best smoothed arm" rather than "best arm". This is arguably a reasonable benchmark when needle-in-haystack functions are allowed. So, this should be mentioned.

In a revision, one should consider the minor comments below, and also address some issues with related work.

MINOR COMMENTS

Does the characterization extend to some particular families of reward distributions, other than binary and gaussian? Perhaps an arbitrary family of reward distributions, under some mild conditions? Some such extension seems feasible, and would be worthwhile to add, in my opinion. (BTW, do mention somewhere that the characterization does hold for binary rewards!)

Discuss the main "learnability condition" (in Thm 1). In particular, note that it doesn't really work for finite #arms (i.e., it always holds), but it does work for countably many arms. Also, non-learnability can be interpreted in terms of "needle-in-haystack" (NIH) intuition: namely, the function $f$ that approaches the inf in (1) up to some eps>0 can be interpreted as an eps-approximate NIH w.r.t. distribution $p$ over arms. Thus, for any $p$ there must exist an eps-approximate NIH.

Does a similar characterization holds for linear vs sublinear regret? This is another standard variant of "learnabiity" in stochastic bandits, hence a natural question to ask and comment on.

In the DEC-like characterization (Sec5), spell out the resemblance to DEC very explicitly (since it seems like a primary motivator for this section), and note that the characterization assumes the existence of a regression oracle (I mean, otherwise the positive and negative results do not quite match).

RELATED WORK

Cite the original papers for the three lines of work discussed in the Intro (note that they considerably predate the papers that you do cite). Specifically, linear bandits trace back to [AK04, MB04], Lipschitz bandits trace back to [KSU08], Bubeck et al (2011) and Slivkins (2011), and structured bandits trace back to [AK11]. For Lipschitz bandits, one should also cite the earlier work on "continuum-armed" bandits [A95, K04]. Likewise, contextual bandits (mentioned in Conclusions) trace back to [LZ07], if not earlier.

Mention that learnability has been well-understood for most/all function classes studied in prior work. E.g., linear bandits are learnable iff the dimension is finite (I think). And Lipschitz bandits are learnable iff the Lipschitz constant is bounded. Further, for a fixed Lipschitz constant, Lipschitz bandits on a given metric space are learnable iff some notion of dimensionality is finite [KSU08].

Mention that there are different "perspectives" one could take when studying structured bandits and/or particular function classes. Aside from learnability, one could optimize the worst-case regret bound: specifically, constant in front of the $\sqrt{T}$, like in linear bandits, or the $\epsilon$ in $T^{\epsilon}$, like in Lipschitz bandits. Also, one could optimize the per-instance regret bound, specifically the constant in front of the $\log(T)$ term, like in Combes et al (2017).

I assume that the prior work on DEC does not imply a similar "learnability characterization". So, this should be stated/explained explicitly.


[A95] Rajeev Agrawal. The continuum-armed bandit problem. SIAM J. Control and Optimization, 1995.

[AKS11] Kareem Amin, Michael Kearns, and Umar Syed. Bandits, query learning, and the haystack dimension. COLT 2011.

[AK04] Baruch Awerbuch and Robert Kleinberg. Online linear optimization and adaptive routing. STOC 2004, JCSS 2008.

[MB04] H. Brendan McMahan and Avrim Blum. Online Geometric Optimization in the Bandit Setting Against an Adaptive Adversary. COLT 2004.

[K04] Robert Kleinberg. Nearly tight bounds for the continuum-armed bandit problem. NIPS 2004.

[KLSZ19] Akshay Krishnamurthy, John Langford, Aleksandrs Slivkins, Chicheng Zhang:
Contextual Bandits with Continuous Actions: Smoothing, Zooming, and Adapting. COLT 2019, JMLR 2020.

[KSU08] Robert Kleinberg, Aleksandrs Slivkins, Eli Upfal:
Multi-armed bandits in metric spaces. STOC 2008, JACM 2019.

[LZ07] John Langford, Tong Zhang: The Epoch-Greedy Algorithm for Multi-armed Bandits with Side Information. NIPS 2007.

POST-REBUTTAL COMMENTS

I appreciate the authors' responses and clarifications.

- I understand that this is the first non-trivial combination of upper/lower bounds that applies generically.

- Good that the results extend to general families of reward distributions and to linear vs sublinear regret.

- Re DEC-like characterization: I see, both upper and lower bounds are stated for a particular regression oracle (in terms of its performance function $\bar{\epsilon}(T)$). So, this should be clarified somewhere.

Re examples: it is interesting to work out both "learnable" and "non-learnable" examples, particularly so for the same "domain". For non-learnability: e.g., infinite $K$ for $K$-armed bandits; infinite $d$ for Lipschitz (resp., linear) bandits on $R^d$; infinite covering dimension (?) for Lipschitz bandits on general metric spaces. I wonder if there are less "obvious" examples of non-learnability that you might be able to work out, especially ones that were not known previously. I think it would improve the paper.

I strongly encourage the authors to revise as per the review / discussion: add examples, extensions, and related work; give more intuition for the main characterization; clarify the the DEC-like result; and point out that "learnability for arbitrary bounded-variance reward distributions" is equivalent to "learnability for binary (resp., Gaussian) reward distributions".

**Paper Award:**

No

---

> ### Author Response · Authors · 2024-11-22
>
> Thank you for your helpful suggestions and feedback! We will take them into consideration in the final revision.
>
> We will add more examples such as: classic multi-arm bandit (set of all bounded functions on a finite set of arms), linear bandits, and Lipschitz bandits to show how our quantity works (see response to reviewer uSop).
>
> Regarding the exponential gap between our upper bound and lower bound, while it is clearly desirable to reduce the gap, we remark that since our work is the first complete characterization of learnability for stochastic bandits, it means our work is the first one to improve the gap between upper bound and lower bound on query complexity from infinite to finite (exponential). The significance of this should be more than any quantitative improvement.
>
> Regarding extending to some particular families of reward distributions, other than binary and Gaussian:
> Yes, we believe it does. Our lower bound proof for Gaussian noise is really quite general, and should apply to most commonly-studied families of distributions (essentially all it relies on is a unified discretization scheme for histogram approximation of the distributions; for instance, this would hold for most bounded families of Lipschitz densities).
>
> -Discuss the main "learnability condition" (in Thm 1). In particular, note that it doesn't really work for finite arms (i.e., it always holds), but it does work for countably many arms. Also, non-learnability can be interpreted in terms of "needle-in-haystack" (NIH) intuition: namely, the function  that approaches the inf in (1) up to some eps>0 can be interpreted as an eps-approximate NIH w.r.t. distribution over arms. Thus, for any  there must exist an eps-approximate NIH.
>
> We suppose by "doesn't really work for finite arms" you just mean that the function class is irrelevant to learnability, since then even the class of all bounded functions is learnable.  We agree, and this is easily implied by our Theorem 1 (where $\gamma \geq 1/K$ for $K$ arms).  Of course, even for finite arms, the result may imply quantitative guarantees beyond this as well.  We will add discussion explaining more about different situations and how theorem 1 applies.
>
> -Does a similar characterization holds for linear vs sublinear regret? This is another standard variant of "learnabiity" in stochastic bandits, hence a natural question to ask and comment on.
>
> Yes, it does. It is straightforward to express reductions (both ways) between query complexity and regret style guarantees.  See theorem 2 in the "Bandit Learnability can be Undecidable" paper, where such reductions are given (showing equivalence of bandit PAC learnability and no-regret learnability). Our proposed complexity measure is therefore also a characterization of learnability in the no-regret setting with arbitrary noise. Given this is a natural question, we will add a new section to state this result in the final version of this paper.
>
> -In the DEC-like characterization (Sec5), spell out the resemblance to DEC very explicitly (since it seems like a primary motivator for this section), and note that the characterization assumes the existence of a regression oracle (I mean, otherwise the positive and negative results do not quite match).
>
> For the dec part, our modification has two parts: 1. We change distribution estimation oracle to regression oracle, since our focus is function classes and the models allow arbitrary zero-mean noise, distribution estimation would not make sense anymore; 2. We change the expectation part $E_{\pi \sim p}[f(\pi^*)-f(\pi)]$ to probability  $P_{\pi \sim p}(f(\pi^*)-f(\pi)>\alpha)$. The second part is essential to guarantee that it characterizes learnability (see response to reviewer uSop), as in our $\gamma$ complexity measure.
>
> Also, we don't need this regression oracle existence assumption. In our dec variant, if we choose $T$ to be 0, the EST term will be 1. Therefore, when the function class does not have a regression oracle, we can choose $T$ to be 0, whereby all the functions in the function class satisfy the estimation condition in the dec term. The dec will degrade to our proposed complexity measure $\gamma_{\mathcal{F},\alpha}$. In other words, our dec variant could improve the query complexity in some cases while still guaranteeing a characterization of learnability without any assumption. This does not contradict the negative result.
>
> Thank you for all the recommended references. We will add them in the final version of this paper.

---

> ### Comment · Reviewer_VKQB · 2024-11-25
> **Post-rebuttal comments**
>
> Thanks for your responses.
>
> I understand your point that this is the first non-trivial combination of upper/lower bounds that applies generically.
>
> Good that your results extend to general families of reward distributions and to linear vs sublinear regret.
>
> Re examples: it is interesting to work out both "learnable" and "non-learnable" examples,  particularly so for the same "domain". For non-learnability: e.g., infinite $K$ for $K$-armed bandits; infinite $d$ for Lipschitz (resp., linear) bandits on $R^d$; infinite covering dimension  (?) for Lipschitz bandits on general metric spaces. I wonder if there are less "obvious" examples of non-learnability that you might be able to work out, especially ones that were not known previously. I think it would improve the paper.
>
> Re DEC-like characterization: I see, both upper and lower bounds are stated for a particular regression oracle (in terms of its performance function $\bar{\epsilon}(T)$). If I understood it correctly, then I suppose the significance of this result not that it is DEC-like, but that it characterizes learnability with a (particular) regression oracle. Either way, this should be clarified.

---

### Official Review · Reviewer_uSop · 2024-11-08
**Review for ALT 2025 Submission 130**

**Rating:** 6
**Confidence:** 3

**Review:**

**Summary**

In this paper, the authors study the stochastic bandits problem in a general setting. They provide a characterization of the learnability of the model class under any zero-mean noise distribution by introducing a new complexity measure, $\gamma_{\mathcal{F}, \alpha}$. They demonstrate that the optimal query complexity of stochastic bandits can be both upper-bounded and lower-bounded by this complexity measure, indicating that it effectively characterizes the complexity of the learning problem. They further extend the measure to accommodate unbounded noise and Gaussian noise, showing that the complexity measure can still characterize the complexity of the classes in more general regimes. Finally, they propose a variant of the decision-estimation coefficient based on $\gamma_{\mathcal{F}, \alpha}$ and prove that it can also characterize the learnability of stochastic bandits.

**Strengths**

- I find it interesting that the authors introduced a new complexity measure for stochastic bandits. They establish both an upper bound and a lower bound for their measure, indicating that their complexity measure can characterize the learnability of the function classes.

- The paper is generally well-written, and the proofs appear to be both non-trivial and correct.

**Weaknesses**

- Although the paper provides a new complexity measure for learnability, the results they derive have a lower bound of $\Omega(\log 1/\gamma)$ but an upper bound of $\mathcal{O}(1/\gamma)$, which creates an exponentially large gap, making it potentially less useful for estimating the optimal sample complexity in general. I understand that the primary goal of this paper is to provide a complexity measure for learnability, but I think the decision-estimation coefficient also provides a way to characterize learnability by measuring the sample complexity. Thus, I would suggest that the authors include the value of $\gamma_{\mathcal{F}, \alpha}$ for common bandit instances, e.g., multi-armed bandits or linear bandits, to show that the proposed complexity measure is computationally feasible in general.

- The authors proposed a variant of the decision-estimation coefficient. However, there is no discussion about what differs from the original decision-estimation coefficient. I would suggest that the authors include a comparison paragraph to improve the presentation of the paper.

- Typo: there is an $\alpha$ missing in the definition of $dec_{\epsilon, \alpha}(\mathcal{F})$ on Page 9. The correct definition should be $dec_{\epsilon, \alpha}(\mathcal{F}) = \sup_{\bar{f} \in \mathrm{co}(\mathcal{F})} dec_{\epsilon, \alpha}(\mathcal{F}, \bar{f})$.


I am willing to increase my score if the authors address my concerns.

**Paper Award:**

No

---

> ### Author Response · Authors · 2024-11-24
>
> Thank you for your feedback!
>
> Regarding the exponential gap between our upper bound and lower bound, while of course reducing the gap is desirable, we remark that since our work is the first complete characterization of learnability for stochastic bandits, it means our work is the first one to improve the gap between upper bound and lower bounds on query complexity from infinite to finite (exponential). The significance of this should be more than any quantitative improvement.
>
> In addition, the previous DEC does not provide a complete characterization of learnability for stochastic bandits. Consider the dec formulation in the paper "Tight Guarantees for Interactive Decision Making with the Decision-Estimation Coefficient".
>
>
> $\textbf{dec} _{\varepsilon}(\mathcal{M})=\sup _{\overline{M} \in \text{co}(\mathcal{M})} \inf _{p,q \in \Delta(\Pi)} \sup _{M \in \mathcal{M}} \\{\mathbb{E} _{\pi \sim p}\left[ f^M(\pi _M)-f^M(\pi) \right]|\mathbb{E} _{\pi \sim q} \left[ D _H^2(M(\pi), \bar{M}(\pi))\right] \leq \varepsilon^2\\}$
>
>
> In the original upper bounds, the value $\varepsilon$ is set based on achievable guarantees for distribution estimation.  Since we are considering arbitrary noise, a highly complex family of distributions is possible, so that it is impossible to achieve non-trivial guarantees $\varepsilon$ for estimating the distributions, and hence the constraint in the definition does not rule out any functions $f^M$.  Given this fact, consider concretely a function class $\mathcal{F}_1=\text{the set of all functions on two arms}$.  The value of $\textbf{dec}$ is $1/2$ (at best $p$ is uniform on the two arms). This case is learnable since the arm number is finite. However, consider another case $\mathcal{F}_2=${$\frac{1}{2}\mathbb{I}_z,z \in \mathbb{R}$}. Again, due to the arbitrary noise, the $\varepsilon$ value for achievable distribution estimation guarantee will not rule out any functions, and then note that for any $p$, there exist some arm $\pi$ where $p$ has 0 probability mass, and there is a model $M$ that has $f^M(\pi)=\frac{1}{2}$. Thus $\textbf{dec}=1/2$ in this case too. This class is not learnable. Since there exists a learnable class and a nonlearnable class that have the same $\textbf{dec}$ value, this indicates $\textbf{dec}$ (at least, as it was used in that work) cannot characterize learnability. In the context of arbitrary noise, there is an inherent drawback in the original definition of $\textbf{dec}$, in that it is based on distribution estimation, which is impossible for arbitrary noise (since it is such a rich class of models).
>
> For common bandit instances, our quantity can be calculated (though where the instance falls in the range between upper and lower bound will vary from instance to instance). For the classic multi-armed bandit (the set of all bounded functions on $K$ arms), choose $p$ in $\gamma$ to be a uniform distribution over each arm, so that $\gamma \geq 1/K$; for linear bandits (with bounded norm), we may construct an $\alpha$-net over the arm space (a bounded subset of $\mathbb{R}^d$). The size of this $\alpha$-net is $(\frac{1}{\alpha})^d$. Letting the distribution $p$ be uniform over those net elements, $\gamma=\Omega(\alpha^d)$; the same $\alpha$-cover idea works for Lipschitz bandits on a bounded subset of $\mathbb{R}^d$, and a similar idea extends this to Holder smoothness.
>
> For our modified $\textbf{dec}$, our modification has two parts: 1. We change the constraint from being based on distribution estimation to a squared loss constraint as can be provided by a regression oracle; since our focus is function classes with models given by arbitrary zero-mean noise, distribution estimation would not make sense anymore; 2. We change the expectation part
> $E_{\pi \sim p}[f(\pi^*)-f(\pi)]$ to probability $P_{\pi \sim p}(f(\pi^*)-f(\pi)>\alpha)$. The second part is inspired by the idea of having a final stage in learning, where samples from $p$ are used to hunt for a good arm, and we only need that some number of samples guarantee one of these arms is good (as opposed to an expectation value, which cannot distinguish between distributions $p$ that might have zero mass on the near-optimal arms).

---

> > ### Comment · Reviewer_VKQB · 2024-11-25
> > **clarifying questions**
> >
> > Your comparison to the (original) DEC seems to focus on some non-standard reward noise, but what if the reward noise is (say) binary?

---

> > > ### Author Response · Authors · 2024-11-27
> > >
> > > In this work, we mainly focus on characterizing learnability of function classes when allowing arbitrary zero-mean noise, which (as we argued in the rebuttal) is not characterized by the original DEC definition.  We additionally found that our characterization of learnability also applies to binary noise and Gaussian noise (and the argument extends to many other types of noise restrictions).  Regarding the special case of binary noise, since distribution estimation is not a problem for binary noise, we find it plausible that the DEC (from the 2023 COLT paper) may also characterize learnability under binary noise.  We will consider discussing this in the final version.  However, the strength of our characterization is that it applies to *both* the common simple noise models (binary, Gaussian, etc.) *and* to the most general noise model (arbitrary zero-mean noise) which makes no assumption of any parametric form or estimability of distributions.

---

> > > > ### Comment · Reviewer_VKQB · 2024-11-27
> > > >
> > > > Thanks for the clarifications!
> > > >
> > > > If it is indeed claimed (or is otherwise obvious) that DEC characterizes learnability with Gaussian noise, this feels like a pretty big omission and needs to be stated clearly when framing your main contributions (i.e., not just in related work).
> > > >
> > > > However, I'm now puzzled. Your lower bound proof (Thm3) seems to imply that your condition is necessary even for binary noise. So, does it follow that "learnability with arbitrary noise" is equivalent to "learnability with binary noise"? But then the DEC condition characterizes both, no? What am I missing?

---

> > > > > ### Author Response · Authors · 2024-11-30
> > > > >
> > > > > In our response, we were perhaps too terse in our remark that "distribution estimation is not a problem for binary noise". We apologize for any confusion. Even with binary or Gaussian noise, the complexity of online distribution estimation can be large. We merely meant that, in this case, the complexity of the online distribution estimation problem is purely dependent on the function class $\mathcal{F}$, *not* on the richness of the noise model. In particular, this means that, in the case of a model class induced by a function class $\mathcal{F}$ combined with binary or Gaussian noise, the upper bound in the 2023 DEC work can be expressed purely based on the function class $\mathcal{F}$, and is not made trivially vacuous by the richness of the noise model. Thus, it is at least *plausible* that it could be useful in expressing a characterization of learnability with binary or Gaussian noise, though it is by no means *obvious* whether it is true or not true. Establishing it would likely require considering the set of all function classes satisfying *our* characterization of learnability, and arguing about whether some appropriate kind of online estimation oracle has a sufficiently small complexity (to use as the epsilon parameter) for all such classes. We will discuss this direction in the final version of the paper.
> > > > >
> > > > > Regarding your second question, since the DEC depends on the *model class*, not just the function class $\mathcal{F}$, we may note that even if the above were to be successful, showing the DEC (with appropriate estimation complexity argument) of the model class "$\mathcal{F}$ plus binary noise" characterizes learnability of $\mathcal{F}$ under binary noise, it would *not* imply that the DEC of the model class "$\mathcal{F}$ plus arbitrary noise" characterizes learnability of $\mathcal{F}$ under arbitrary noise: the value of the DEC of "$\mathcal{F}$ plus arbitrary noise" is often much larger than the value of the DEC of "$\mathcal{F}$ plus binary noise".  As we have argued, the DEC of the model class "$\mathcal{F}$ plus arbitrary noise" *does not* characterize learnability of $\mathcal{F}$ under arbitrary noise. It is nonetheless an interesting question whether the DEC under one model class ("$\mathcal{F}$ plus binary noise") might offer a characterization of learnability under a different model class ("$\mathcal{F}$ plus arbitrary noise"), as would be implied (by our results) if the above question about DEC and learnability with binary noise could be verified; we will mention that this would be a further implication, in our discussion of the above mentioned direction.

---

> > > > > > ### Comment · Reviewer_VKQB · 2024-11-30
> > > > > >
> > > > > > Got it re DEC, thanks!
> > > > > >
> > > > > > But could you please clarify re the "non-DEC" part of my question:
> > > > > >
> > > > > > Your lower bound proof (Thm3) seems to imply that your condition is necessary even for binary noise. So, does it follow that "learnability with arbitrary noise" is equivalent to "learnability with binary noise"?

---

> ### Author Response · Authors · 2024-12-01
>
> Yes, our results imply that for any function class $\mathcal{F}$, learnability of $\mathcal{F}$ with arbitrary noise is equivalent to learnability of $\mathcal{F}$ with binary noise (and equivalent to learnability with Gaussian noise).

---

### Author Rebuttal · Authors · 2024-11-25

We thank all of the reviewers for their encouraging remarks and helpful comments and feedback. We respond to reviewers individually in separate comments to each reviewer.

---

### Meta-Review · Area_Chair_jVbV · 2024-12-13

**Recommendation:** Accept
**Confidence:** 3

**Metareview:**

This paper studies learnability for general bandits with stochastic observations, given a realizable function class mapping arms to expected rewards. It provides a characterization of learnability by introducing a new complexity measure and showing that the query complexity to learn a near-optimal policy with high probability is within an exponential band of this complexity measure. This applies to bounded zero mean noise distributions as well as ones with bounded variance. The paper further provides an alternate characterization when a regression oracle is available by proposing a variant of the DEC.

All reviewers appreciated the generality of the main result, which answers an open question posed by Hanneke and Yang (2023). The paper is also generally well written but sufficient discussion, examples and context is missing in several places. This caused concerns on the significance over existing work. The authors' rebuttals and discussion could clarify the significance and how the DEC variant introduced here relates to the existing one.

All in all, while the reviewers do have some hesitations, they are leaning positively. After the clarifications in the author-reviewer discussion, I tend to agree with the reviewers' positive sentiments and recommend acceptance. However, the authors should carefully revise the paper to incorporate the feedback from the reviewers. The most salient points to address are:

* Provide more intuition for the main characterization
* Add examples for what the complexity measure evaluates to in special cases, ideally both positive and negative examples for learnability
* Provide a careful discussion of how the proposed DEC variant relates to prior variants,  in the definition and their capability to characterize learnability in the general setting and special cases.

Also, if possible, adding extensions to regret minimization and more general reward distributions would further strengthen the paper.

**Paper Award:**

No